# Nervous Necrosis Virus Modulation of European Sea Bass (*Dicentrarchus labrax*, L.) Immune Genes and Transcriptome towards Establishment of Virus Carrier State

**DOI:** 10.3390/ijms242316613

**Published:** 2023-11-22

**Authors:** Dimitra K. Toubanaki, Antonia Efstathiou, Odysseas-Panagiotis Tzortzatos, Michail-Aggelos Valsamidis, Leonidas Papaharisis, Vasileios Bakopoulos, Evdokia Karagouni

**Affiliations:** 1Immunology of Infection Group, Department of Microbiology, Hellenic Pasteur Institute, 11521 Athens, Greece; dtouban@pasteur.gr (D.K.T.); toniaef@pasteur.gr (A.E.); ptzortzatos@pasteur.gr (O.-P.T.); 2Department of Marine Sciences, School of the Environment, University of the Aegean, University Hill, Lesvos, 81100 Mytilene, Greece; mard16008@marine.aegean.gr (M.-A.V.); v.bakopoulos@aegean.gr (V.B.); 3Avramar S.A., 19.3rd Km Markopoulo-Paianias Av., 19002 Paiania, Greece; l.papaharisis@avramar.eu

**Keywords:** viral nervous necrosis, nervous necrosis virus, nodavirus, European sea bass, persistent infection, carrier state, transcriptome, immune genes, host–pathogen interaction

## Abstract

Viral infections of teleost fish have great environmental and economic implications in aquaculture. Nervous necrosis virus (NNV) is a pathogen affecting more than 120 different species, causing high mortality and morbidity. Herein, we studied the course of NNV experimental infection of *D. labrax*, focusing on survivors which indicated viral carrier state. To determine the carrier state of *D. labrax* head kidney, we performed a gene expression analysis of selected immune-related genes and we profiled its transcriptome 14 days post infection (dpi). All tested genes showed clear differentiations in expression levels while most of them were up-regulated 14 dpi suggesting that their role is not limited in early antiviral responses, but they are also implicated in disease persistence. To gain a better understanding of the fish that survived the acute infection but still maintained a high viral load, we studied the differential expression of 124 up-regulated and 48 down-regulated genes in *D. labrax* head kidney, at 14 dpi. Concluding, the NNV virus persistent profile was assessed in *D. labrax*, where immune-related gene modification was intense (14 dpi) and the head kidney transcriptome profile at this time point offered a glimpse into host attempts to control the infection in asymptomatic carriers.

## 1. Introduction

Viral infections of teleost fish have great environmental and economic implications in aquaculture; therefore, the related host–pathogen interactions are under intense investigation [1]. Aquaculture holds a major part of the global food system, providing over half of the fish production worldwide; therefore, viral outbreaks in farmed species have great environmental and economic implications [2]. The onset of an infectious disease in a specific host depends both on pathogen virulence and the host immune response [3] and can be assessed in vivo by analyzing immune-related genes expression. Equally important in disease progression seems to be the viral carrier state mechanisms; however, the understanding of the ways in which the host’s immune response can cope with the virus presence is still limited.

Nervous necrosis virus (NNV) is a pathogen affecting more than 120 different species from marine and freshwater environments, causing high mortality and morbidity of cultured species [4,5,6]. NNV is part of the genus Betanodavirus and is a member of the Nodaviridae family, with a genome consisting of two positive-sense ssRNA molecules (RNA1 and RNA2) in a non-enveloped capsid. Betanodaviruses are classified into four genotypes, i.e., striped jack NNV (SJNNV), red-spotted grouper NNV (RGNNV), tiger puffer NNV (TPNNV), and barfin flounder NNV (BFNNV) [7]. NNV is the causing agent of viral nervous necrosis (VNN) or vacuolating encephalopathy and retinopathy (VER) or encephalomyelitis, a disease resulting in mortality rates of up to 100% in some species. The clinical symptoms of NNV infection depend on the fish species, biological stage, phase of disease and temperature and include abnormal swimming behavior (spiral swimming, whirling, horizontal looping or darting), loss of appetite, swim bladder hyperinflation and coloration abnormalities, resulting eventually in the death of infected hosts [5,6,8,9]. European sea bass (*Dicentrarchus labrax*), one of the major hosts of NNV, is a teleost fish found in the Mediterranean, north-eastern Atlantic Ocean and Black Sea; it is well characterized and economically important since it is ranked fourth in European aquaculture production [10].

Sea bass cellular and molecular immune responses to NNV infection have been evaluated in several studies [3,11,12,13,14,15,16,17,18]; however, they are not definitely characterized, especially in a long-term period following infection (i.e., more than 10 days). The host responses to NNV infection have been associated with differences in expression of interferon response markers, pro-inflammatory genes and transcription factors. As observed in all vertebrate classes, the main innate defense mechanisms of antiviral response include the production of interferon type I (IFN) peptides and the Mx protein, deregulation of genes involved in innate responses like hepcidin, transferrin, cxcr4, and apolipoprotein, and potent upregulation of the IgM light chain [12]. The signal transducer and activator of transcription (STAT1) is involved in the innate immune response against NNV infection [19], while tumor necrosis factor alpha (TNFα) and interleukin-1 beta (IL-1β) seem to be upregulated when NNV replicates in its target organ [11]. Moreover, cluster of differentiation 8 alpha (CD8α) was found increased in NNV-infected fish, revealing that NNV also induces specific immune responses [20].

These studies have focused on the early activation of the host’s response to NNV immediately after infection. However, some studies have also shown that interferon is involved in the persistence of NNV infection [21,22]. Viral persistence, a term often used to describe viral carrier states, is not well studied in fish. The evidence of persistence has been found for a few fish viruses, including infectious pancreatic necrosis virus (IPNV) [23], NNV [24], koi herpesvirus (KHV) [25], viral hemorrhagic septicemia virus (VHSV) [26] and infectious hematopoietic necrosis virus (IHNV) [27], providing some information of how viral persistence is developed and maintained. Regarding NNV persistent infection, it has been observed that the symptoms of persistent infection are different from those of typical acute infection, i.e., the diseased fish show no clinical signs of VNN but virus particles are detectable in fish brain and retina [28]. The transition from acute to persistent infection is the key stage to understand the mechanisms leading to carrier state with respect to developing improved disease management practices.

Thus, the aim of the present study was to evaluate the modulation of immune-related genes in European sea bass, challenged with Nodavirus, in specific time points, in a long-term study. Fish physiology parameters were recorded and analyzed to assess the infection effects on the fish health. The kinetics of key genes associated with the immune response proved to be useful to understand the ability of fish to activate immune protective-related mechanisms on VNN disease onset from a system point of view and offered a glimpse on host mechanisms responsible for NNV infection progress towards the establishment of the virus carrier state in sea bass. In an attempt to further understand the transition to the virus carrier state, the transcriptome differences between head kidney tissue obtained from naïve (no virus exposure) and carrier fish (exposed to NNV and tested positive for NNV in the brain) were profiled in a late time point (14 days post infection).

## 2. Results

### 2.1. Cumulative Survival of D. labrax Challenged with NNV Reached 79% with Mortality Peak on Days 5–6 

An experimental NNV challenge via intramuscular injection was utilized to study NNV infection progress in 28 days post infection period. A summary of the experimental setup is given in Figure 1. 

To evaluate the effects of NNV infection on fish viability, mortality rate was recorded daily throughout the study. Typical signs of VNN were observed in fish from Day 3. The cumulative survival rates for each group are shown in Figure 2A, where the highest daily mortalities are also highlighted. The cumulative survival rates in each experimental tank with NNV-infected fish are also presented in Appendix A, where the absence of significant variations in fish mortality between tanks is confirmed. No mortalities were registered in non-infected sea bass, whose survival remained 100% up to the end of the experiment. By contrast, a gradual decline in infected fish was observed from Day 1 post infection (99.57% survival) to the end of the experiment, i.e., Day 28 (79.17% survival). As shown in Figure 2A, the daily mortalities reached a peak on Days 5–6 and continued to be high on Days 8–12 (as indicated by arrows), reaching over 83% of the total mortality incurred during the experiment (i.e., 40 fish of 48 fish in total died up to Day 12). After this 12-day period, mortality greatly subsided with only eight deaths occurring over the subsequent 16 days.

### 2.2. NNV Viral Load Remained High until the End of Experimentation

Viral RNA1 present in sea bass brain was quantified by absolute real-time PCR, which revealed the presence of NNV in challenged fish 3 days post infection (dpi) (Figure 2B). The highest levels of viral load were observed on 7 dpi (54.3 ± 2.2 × 10^11^ TCID_50_ of the NNV per µg of total RNA). Viral load was then decreased at 14 dpi, which might be explained by the mortality observed around 7 dpi, likely caused by the removal of fish with the highest viral load. The viral load at the final time point (28 dpi) slightly increased while mortality ceased, suggesting that the surviving fish could be asymptomatic carriers with high viral loads almost one month following the initial infection. Despite the presence of the virus in the brain, no phenotypic or behavioral changes were observed in infected fish and no natural fish loss occurred during the experiment. No viral load was observed in the non-infected group.

### 2.3. Biometrical Data Were Not Significantly Affected by NNV Infection

No statistically significant differences were obtained in specimen weights between the infected and the control groups (Appendix A). Fish weights were unaltered in the non-infected group in the 1–28-day period whereas the weights changed significantly in the VNN-infected group between the first two days of the experimental challenge and Day 28.

Three parameters related to fish weight, i.e., weight gain, specific growth rate and condition factor, were calculated to assess the NNV infection effects in more detail (Appendix A). Both experimental groups seemed to lose weight the first two days of experimentation, possibly due to the experimental handling and the injection shock (either PBS or the virus), as depicted very clear by the weight gain and the specific growth rate parameters which appear negative at 24 and 48 hpi. On Day 3, the non-infected group weight gain was slightly decreased and subsequently increased gradually until the end of the experimentation, reaching ~30% of weight gain. On the contrary, the NNV-infected group appeared to gain weight more quickly on 3 and 4 dpi. On 7 dpi, weight gain dropped sharply, fish seemed to regain weight on 14 dpi and the weight gain ended up at ~40% at 28 dpi (Appendix A). Statistically significant differences on the weight gain factor were recorded when 28 dpi were compared to 0, 24 hpi, 48 hpi and 7 dpi. As seen in Appendix A, the specific growth rate has great variation for both groups in the initial time points (24 hpi–3 dpi); however, it is stabilized on the following time points. Finally, the condition factor parameter seems unchanged, both for the control and the NNV-infected groups, throughout the experimentation (Appendix A).

Then, splenosomatic index (SSI) calculations resulted in statistically significant differences between the NNV-infected and non-infected groups on 6 hpi (0.25 dpi) and 4 dpi (Appendix A). Assessment of the SSI at each studied time point for the control group revealed a statistically important difference on 24 hpi (1 day) when compared with 4 dpi. On the contrary, more important differences were recorded for the NNV-infected group as expected: the SSI increased significantly between 6 and 24 hpi. The SSI was subsequently decreased on 7 dpi, at a level significantly lower than that in the previous time points.

### 2.4. Haematological Parameters Were Significantly Affected by NNV Infection

Haematocrit (Hct) values reveal statistically important differences among the NNV-infected and control groups at the early time points, i.e., 3 and 6 hpi (Appendix A). Hct appears high (>40% of total blood volume) at a very early time point (3 hpi), rapidly decreases to half at the next time point (6 hpi) and remains stable throughout the experimentation (~20% of total blood volume). Statistically significant differences in the non-infected group are recorded between 3 hpi and 24 hpi, 48 hpi and 3 dpi. The NNV-infected group results in important differences between 3 hpi and all other time points except 24 hpi, 7 dpi and 28 dpi. A significant difference is found in leucocrit (Lct) values among NNV-infected and non-infected groups at 24 hpi, 48 hpi and 7 dpi (Appendix A). In general, Lct is higher at 12 hpi for both groups than at all other time points, including 6 hpi. Statistically significant differences in the control group are recorded when 7 dpi are compared to 6 and 12 hpi, between 12 and 48 hpi, as well as between 3 and 7 dpi. The NNV-infected group reveals more important differences at the tested time points: 14 dpi has a lowered Lct compared to 6, 12 and 24 hpi. Likewise, Lct is further lowered at 28 dpi, while 4 and 7 dpi are significantly decreased compared to 12 hpi.

### 2.5. Gene Expression of D. labrax Challenged with NNV

The gene expression profile was analyzed to evaluate the time-dependent modulation of immune-related genes in NNV-infected and non-infected sea bass. Of the 15 fish sampled per time point for infected group, 9 fish were employed on gene expression studies, namely those who had adequate total RNA with the best quality for cDNA preparation. For the non-infected control group, all five sampled fish per time point were used on gene expression studies.

#### 2.5.1. Interferon Pathway Related Genes Were Differentiated at Early Time Points and 14 dpi

The expression profiles of interferon pathway-related genes are presented in Figure 3. Antiviral protein MxA had no statistical differences in expression levels at early time points (<3 dpi). The MxA expression levels increased 3 hpi (1.5-fold), and a significant increment was observed at 7 dpi (4.5-fold). The mean expression of the gene remained higher than in the early time points on 14 dpi (2.7-fold) and was lowered on 28 dpi. IRF7 expression levels were significantly decreased on 6 hpi (0.5-fold), followed by an increment which reached the initial levels on 24 hpi. A second cycle of decrement was observed until 4 dpi; IRF7 expression levels were increased 7 and 14 dpi (1.4-fold) and were finally kept at low levels on 28 dpi. ISG12 expression levels increased on 6 and 24 hpi (>3-fold), with further increment on 4 and 7 dpi (>5-fold). An impressive statistically significant increase compared with all time points up to 4 dpi was observed on 14 dpi (77-fold), followed by a decrement on the basal level on 28 dpi.

#### 2.5.2. Cytokine Gene Expression Was Affected by NNV Infection Mostly in Late Time Points 

Figure 4 contains the expression profiles of four cytokine-related genes. The expression levels of IL-1b were high at the initial time point (0 hpi) and were lowered at all subsequent time points up to 7 dpi. The IL-1b expression increased significantly on 14 dpi (5.2-fold), followed by a decrease to basal levels at the final time point (28 dpi). IL-10 appeared to have variable expression levels: it was increased at the initial time point (0–3 hpi) and drastically decreased on 6 hpi. At the following time points (12 hpi–4 dpi), the IL-10 expression levels remained low and were increased on 7 dpi (1.7-fold). At the final time points (14 and 28 dpi), IL-10 gradually returned to low expression levels. STAT3 appeared to have unaltered expression levels in all time points up to 14 dpi, where a significant increase (3.3-fold) was observed. On 28 dpi, STAT3 expression decreased at the basic levels. On 3 hpi, TNFa expression was slightly increased (1.5-fold) whereas at all subsequent time points the gene expression was decreased at various levels. The lower expression was observed at the final time point (28 dpi).

#### 2.5.3. T-cell Marker Gene Expression Levels Were Fluctuated at Early Time Points but Were Up-Regulated at 14 dpi

CD4 gene expression decreased significantly 6 hpi, increased to the basic level until 24 hpi and significantly increased 14 dpi (1.6-fold), followed by a return to the initial levels at the final time point (28 dpi) (Figure 5). CD8a expression levels increased significantly 12 hpi (3.4-fold), followed by gradual decrement up to 3 dpi. An increment was observed at 4 dpi (3.5-fold), expression levels were lowered on 7 dpi and increased again on 14 dpi (4.4-fold). At the final time point, i.e., 28 dpi, CD8a expression reached its lower point (Figure 5).

#### 2.5.4. Antimicrobial Peptide Gene Expression Was Not Significantly Affected by NNV Infection

Hepcidin (Hep) expression levels were increased at the initial time point (0 hpi) and slightly decreased on very early points (3–6 hpi). Hep expression reached a maximum on 24 hpi (6.5-fold), gradually decreased at the following time points and remained low until the end of experimentation (Figure 6).

#### 2.5.5. Immunoglobulin Analysis Revealed an Increasing Trend to Late Time Points

IgHM gene expression results are presented in Figure 7A. IgHM expression increased at 6–12 hpi, slightly decreased at the following time points and began to increase again at 4 dpi. IgHM expression increased significantly at 14 dpi (16-fold) and finally at 28 dpi the IgHM levels appeared lower but highly above the basal level. Specific anti-nodavirus antibody levels fluctuated insignificantly below the threshold levels the first 4 days post infection and they gradually increased from 7 to 28 dpi (Figure 7B). Specific anti-nodavirus Ab levels of all time-points from 0 to 7 dpi were significantly lower compared to those of 14, 21 and 28 dpi. 

### 2.6. Transcriptome Analysis of NNV-Challenged D. labrax Head Kidney 14 dpi 

Head kidney total RNA isolated from NNV-infected (*n* = 5) and non-infected (*n* = 4) fish (RIN ≥ 7.5) were subjected to sequencing. Sequencing of the nine libraries (NNV-infected and non-infected head kidney) yielded a total of 107,140,300 and 61,818,643 sequences for NNV-infected and non-infected samples, respectively (Appendix A). Raw data reads are available in the NCBI database (Accession No. PRJNA1030357). All reads were aligned to a reference genome in a high rate (88.8–96.21%).

#### Differential Expression Analysis and GO Classification Revealed 133 Up- and 68 Down-Regulated Genes with Various Biological Functions

Differentially expressed transcripts were analyzed using the Bioconductor package DESeq2 and were identified according to log2FC ≥ 1 and a *p*-value threshold of less than 0.05. When NNV-infected samples were compared to non-infected samples, 751 statistically significant genes were found, of which 133 genes were up-regulated, 68 genes were down-regulated and 550 were not differentially expressed (Figure 8A). The heat map of two-dimensional hierarchical clustering of the resulting DEGs is shown in Figure 8B. The top 20 highest and lowest differentially expressed genes at 14 dpi are shown in Table 1 and Table 2.

When functionally analyzed using OmicsBox software (version 3.0.30), a total of 124 up-regulated and 48 down-regulated genes could be blasted, GO mapped and annotated from the differentially expressed transcripts on Day 14. The up-regulated and down-regulated genes were classified according to the GO database into the categories of biological process, molecular function and cellular component. For the up-regulated genes (Figure 9A), “cellular process” (36 genes) was the largest subcategory in the category of biological process, followed by “metabolic process” (21 genes). The major subcategory for molecular function was “binding” (41 genes) followed by “catalytic activity” (16 genes). The “cellular anatomical entity” (58 genes) was the major GO term in the cellular component category. For the down-regulated genes (Figure 9B), “cellular process” (25 genes) was the largest subcategory in the category of biological process, followed by “metabolic process” (16 genes). The major subcategory for molecular function was “binding” (19 genes) followed by “catalytic activity” (14 genes). The “cellular anatomical entity” (23 genes) was the major GO term in the cellular component category.

Τhe top 20 differentially expressed genes of each category (up- or down-regulated) were searched against the literature for potential involvement in viral infection. To gain further insights into disease pathogenesis, the orthologs of the differentially expressed genes in zebrafish were manually determined. Of the 140 proteins, 105 were found in the Search Tool for the Retrieval of Interacting Genes/Proteins database of the STRING database and analyzed in terms of network functional enrichments. Oxidative phosphorylation emerged as the enriched KEGG pathway (5/135-red nodes) (Figure 10). 

## 3. Discussion

Betanodavirus is an RNA virus that infects the brain and the central nervous system causing fatal disease. If the host survives the acute phase of infection, the virus persists resulting in the establishment of asymptomatic carriers [29]. However, the mechanisms by which asymptomatic, chronic subclinical infections (i.e., carrier states) of nodavirus are established, maintained and affect the systemic responses of several fish species are poorly understood.

None of the fish tested prior to the experimental infection were positive for NNV. The described experimental design was carefully chosen in order to take into account individual fish differences as well as possible differences between the experimental tanks. Virus RNA1 was detected 3 dpi, the brain viral load reached its maximum 7 dpi, was slightly lower 14 dpi and appeared high at the final time point (28 dpi). Affected fish showed the typical symptoms of VNN such as swimming disorders [4,6]. Our results agree with previous studies showing that NNV displays fast kinetics of replication [15]. In most studies, viral load could be recorded as early as 2 dpi; however, their methodology is based on qPCRs targeting the virus RNA2 segment. RNA2 (expressing the viral coat protein) is 2–10 times more abundant than RNA1 (expressing the RNA-dependent RNA polymerase) [30]; therefore, a small delay at an initial detection point is expected in our study. In addition, the fast replication scenario agrees with the appearance of the first mortalities that were recorded from the first days of the infection [6]. As mentioned above, the maximum viral load was reached at 7 dpi, which is in accordance with another independent experimentation where fish inoculated with a virus strain causing similar mortalities (~20%) had a significant increase in the viral copy number on the brain at 7 dpi [3]. The viral load at the final time points (14–28 dpi) was still high, while mortality was minimized. Experimental infection of SSN-1 cell cultures with brain homogenates obtained from 14- and 28-dpi samples resulted in extended cytopathic effect, confirming that the surviving fish were asymptomatic carriers with high viral loads almost one month following the initial infection.

Qualitative and quantitative variations in hematological parameters are significant in regard to fish diseases diagnosis [31]. Τhe amount of leukocyte cells is normally lower in healthy fish and can be used as an indicator for infectious diseases. When infectious disease agents enter the fish body, the non-specific (cellular) defense system becomes stimulated during the first stage of disease manifestation. In this situation, leucocytes become increased (leucocytosis) initially in order to protect the fish body by phagocytosis [32]. In the present study, the significant increase found in leucocrit values of NNV-infected compared to those of non-infected groups at 24 and 48 hpi signifies the fact that the fish innate immunity was stimulated to fight the virus as the primary line of defense.

The infection of European sea bass by NNV provokes significant immune responses, which were recently reviewed [6,7]. When NNV infects a fish, the virus spreads to nervous tissues and the brain, as well as to systemic organs, i.e., head kidney, spleen, etc. In general, viral nucleic acids are detected by host cells by pattern recognition receptors (PRRs) which include retinoic acid-inducible gene-I-like receptors (RIG1-like receptors, RLRs). In fish, RLRs facilitate interferon regulatory factor IRF3 and IRF7 activation and their translocation into the nucleus for the induction of interferon IFN-I [1]. Immune responses against NNV infections include the expression of IRF3, interleukins (IL-1b, IL-8, IL-10), tumor necrosis factor (TNF-a), and immunoglobulins (IgM) [33]. Betanodavirus is an intracellular pathogen that causes apoptosis in its host [34], and it is likely that both B cell and cytotoxic T-cell activities are needed to provide an effective adaptive response. Presentation of antigens through an intracellular route induces MHC-I-restricted CD8^+^ responses, while extracellular delivery induces MHC-II-restricted CD4^+^ responses [35]. Finally, antibody activation is an important host immune response during NNV infections in teleosts since the produced antibodies can neutralize the virus preventing it from causing damage [36]. 

The genes included in the present study were designed based on literature reports for the NNV challenge of various fish species to offer a glimpse of immune-related molecule modulation. These genes are members of the interferon pathway, cytokines, transcription factor, T-cell markers; they express an antimicrobial peptide. Immunoglobulin M expression levels were also studied [3,11,12,13,15,37,38]. The obtained results showed clear differentiations in gene expression levels confirming the efficacy of in vivo challenge with NNV, which proved to be able to induce antiviral responses. The experimental setup covers a long time period (0–28 dpi) in order to analyze the progress of the infection from the acute phase up to chronic infection and/or the long-term carrier state. The head kidney was chosen as a target tissue for expression analysis since it is considered ideal for studying the elicited immune responses following an infection. Indeed, most of the studies published on the fish immune response to NNV have focused on the head kidney [39].

Innate immunity represents the first antiviral defense in vertebrates and is mediated by interferon (IFN) and interferon-induced genes (ISGs). Types I and II IFN are present in fish [40] and both have been detected in individuals infected with betanodavirus [8,41]. Type I interferons (IFN) are important defense molecules, generating an intracellular environment that restricts viral replication, as well as regulating the adaptive immune response [29]. Two of the most important ISGs, i.e., interferon-inducible Mx protein A (MxA) and ISG12 ubiquitin-like modifier (ISG12), along with the interferon regulatory factor 7 (IRF7), were assessed in the present study. 

Mx proteins are key components of the antiviral state triggered by IFN type I in response to infections. Two different Mx genes which have antiviral responses with different intensities and spatial and temporal patterns have been identified in European sea bass: MxA and MxB [42]. MxA is the predominant isoform expressed during the VNN infection, and it was chosen as an IFN-I system stimulation marker. In sea bass head kidney, Mx mRNA has been detected at early time points, at times ranging from 6 to 72 hpi, reaching maximum transcription levels 6, 12 or 24 hpi [3,11,12,13,15,42]. Members of the IRF family seem to be important in the immune response of neurological tissues of higher vertebrates to RNA viruses [29], and in teleosts, IRF7 has been identified as a primary transcription factor priming type I IFN response. A study on IRF7 modulation by viral infections in Asian sea bass demonstrated that high transcript abundance was observed 24 h post infection in the brain, heart, kidney and spleen [43] and irf7 was highly expressed 3 dpi in sea bass correlated with the transcription of mxA and isg12 [3]. ISG-12 is a small protein involved in the intrinsic apoptosis pathway activation, and its antiviral activity has been demonstrated for viruses affecting high vertebrates [17]. RGNNV infection induces high levels of the ISG-12 protein in the sea bass comparable to those of Mx and ISG-15. In the head kidney, ISG12 expression levels reached a maximum level 24 hpi and decreased at the following time points (72 hpi) [3,15]. In the present study, ISG12 expression levels appear to have increments in the early time points as expected; however, all three genes (Mx, IRF7, ISG12) are surprisingly up-regulated 7–14 dpi, suggesting that their role is not limited in early antiviral responses, but they are also implicated in disease persistence.

As a pro-inflammatory cytokine, TNF-a is expressed at an early stage of infection in fish and has a key role in regulating inflammation. In the present study, TNF-a was up-regulated at 3 hpi followed by a down-regulation at 6 hpi, which persisted throughout the experimentation. Thus, early inflammation seems to contribute to viral infection control; however, TNF-a levels are reduced in all later time points of the NNV infection. TNF-a activates macrophages/phagocytes and enhances their microbial killing activity. Fish TNF-a is involved in leucocyte homing regulation, proliferation and migration, and exerts pro-apoptotic activity [44] based on its direct effect on NF-kB or apoptosis, depending on the cellular context [45]. TNF-a expression levels after NNV infection has been extensively studied in sea bass. Moreno et al. [15] observed that TNF-α mRNA levels decreased at 12 hpi, with a clear up-regulation at 24 hpi in the head kidney in a 3–72 hpi study. 

The inflammatory response is a cascade triggered by TNF-a, which subsequently leads to the downstream transcription of other pro-inflammatory genes, including IL-1b. In fact, fish TNF-a displays overlapping functions with IL-1b, which are tightly regulated, mainly at a transcriptional level, by anti-inflammatory cytokines [3,15,44]. IL-1b is produced by a wide range of cell types early after the activation of host PRRs, increasing phagocyte migration, macrophage activity, and lymphocyte activation. It also induces the expression of immune suppressive cytokines such as IL-10 and prostaglandins [44]. To evaluate whether the gene expression of pro-inflammatory cytokine IL-1b is correlated with any antiviral activity, its levels were assessed in all time points. Its initial levels were high in the challenged group compared to those of the control group. A marked increase was observed at 14 dpi, revealing extended cytokine expression towards establishing the carrier state. It has been reported that IL-1b displayed a nine-fold increase at 3 dpi with lower expression levels at 24 hpi and 10 dpi time points when infected head kidneys were analyzed [12]. Overall, IL-1b is one of the most important and potent pro-inflammatory cytokines by its early induction against antigens. It enables the host to respond to infection by inducing a cascade of reactions leading to inflammation [45]. The present study suggests the IL-1b potentiality to act protectively for the host towards establishing the virus carrier state (in a long-term infection), a role which needs further elucidation.

The anti-inflammatory cytokine IL-10 acts as pro-inflammatory cytokine suppressor and exerts a conserved role in dampening inflammatory responses [44]. It regulates cytokine expression, immune cell proliferation, and plays a key role in preventing massive inflammatory responses, which may cause tissue damage [46]. Previous studies reported that in sea bass, IL-10 was induced 24 h and 10 days after infection, with a low-level increase at 3 dpi [12], or was increased at 24 and 72 hpi (3 dpi) without any differences at longer times points [15]. The expression of IL-10 showed a high level of variation throughout the present experimentation: it was slightly up-regulated at the initial time points, down-regulated from 6 hpi to 4 dpi, but its expression increased at 7 dpi and again decreased at the final time points. Our results show that repeated cycles of gene up-regulation and down-regulation are indicative of the head kidney effort to balance inflammation which is essential to control NNV replication, avoiding, in parallel, an uncontrolled process which could cause tissue damage. Hence, our results suggest that both the pro-inflammatory response and its regulatory mechanism are increasing in late time points, indicating the host effort to overcome/avoid NNV infection and preserve a carrier state, since it seems that the virus is not cleared from the previously infected fish.

In mammals, IFN-γ is able to activate transcriptional factors such as STAT3, STAT1 complex and IRF9, and is has been suggested that STAT1/3-involved signal transduction of type II IFN is likely to be conserved from teleost to human [40]. STAT3 phosphorylation and nuclear translocation is also induced by IL-10 [44]. In the present study, STAT3 appears up-regulated at 14 dpi while it seems unaltered by infection in all other time points. It has been reported in mammals that the activation of Stat3 leads to a transcriptional response resulting in increased concentrations of SOCS3, which dampens signal transduction by TLRs and cytokine receptors, resulting in decreased inflammatory cytokines and hepcidin transcription [47]. An analogous mechanism seems to be activated in NNV-persistent infection in sea bass.

Overall, the genes encoding cytokines, which are usually up-regulated in response to viral infection, were not differentially modulated in our study, with a striking exception at 14 dpi. These findings, together with the interferon-related gene expression profiles, led us to consider that in this work, we are describing persistence establishing mechanisms that enable *D. labrax* to survive as an NNV carrier.

The T cell-specific genes CD4 and CD8a have been previously analyzed in the NNV infection with controversial results. Scapigliati et al. [12] reported that the T cell-specific genes CD4 and CD8a did not show any significant transcriptional variations in their expression in all tested time points (24 hpi–10 dpi). However, Valero et al. [39] found that NNV infection increased major markers for cytotoxic (cd8a) and helper (cd4) T cells up to 6 dpi (144 hpi). The present study shed some light on late time point expression for both genes, since the early time points did not show significant up- or down-regulations. On the contrary, both genes were significantly up-regulated at 14 dpi. This observation is in accordance with studies on mice [48]; however, the findings are preliminary, and a more detailed analysis should be performed to further elucidate the underlying mechanism.

Hepcidin is a liver-produced hormone that constitutes the main circulating regulator of iron absorption and distribution across tissues and cells, and it has been associated with antifungal and antibacterial activity through binding to cell walls [46]. As mentioned above, in mammals, hepcidin can also act as an anti-inflammatory agent, inducing a signal cascade by hepcidin-activated Jak2, which phosphorylates transcription factor Stat3, subsequently provoking an anti-inflammatory transcriptional response by negative feedback [47]. Even though hepcidin was found to be significantly upregulated in bacterial infections [46], in the present study, it was slightly up-regulated in response to viral infection, confirming previous observations [37]. Finally, teleost B lymphocytes primarily present immunoglobulin (Ig) of the IgM class [49]. The IgM heavy chain gene showed a potent expression soon after infection (6 hpi) in accordance with previous reports [12,39], and a 16-fold up-regulation on Day 14 after infection. The antibody levels in fish plasma, however, were only slightly increased above threshold at 14 dpi. It should be noted that the present study was focused on immune-related gene expression analysis, which offers a time shot depiction at the transcription level. The expressed protein levels may be different, as we observe in Figure 7 for IgHM. Therefore, the assessment of the respective protein levels in a future study will be beneficial for in = depth immune response understanding.

To gain a better understanding of the fish that survived the acute infection but still maintained a high viral load, we studied the differential expression of 124 up-regulated and 48 down-regulated genes in *D. labrax* head kidney, 14 days post NNV infection, since an enhanced immune activity was recorded at that time point. The main GO categories for both up- and down-regulated genes were the cellular process for biological processes (BP), binding for molecular functions (MF) and the cellular anatomical entity for cellular components (CC). Immune system process as a BP subcategory was only found in up-regulated genes.

The up-regulation of a number of genes implicated in virus infection was revealed from the transcriptome analysis. The most highly expressed gene was annotated as tripartite motif (TRIM)-containing protein 16-like (TRIM16L). In mammals, TRIM proteins are one of the most important sets of antiviral effectors and regulators of antiviral defence through an astonishing diversity of mechanisms, from direct viral restriction to modulation of immune signaling and autophagy. In fish, their role remains poorly understood. TRIMs typically possess a ubiquitin (Ub) E3 ligase activity [50]. Fish TRIM orthologs have been found induced by NNV in cod [51], with moderate responses indicating that trim genes participate in the innate antiviral response by regulating the IFN signaling pathways, not having direct effector functions. Recently, a study on orange spotted grouper proved that TRIM16L is localized in the cytoplasm and exerts critical roles on virus replication with different regulatory effects on innate immune response. In fact, EcTRIM16L inhibited the expression of interferon-related signaling molecules and effectors (IRF3, IRF7, IFP35) and negatively regulated MDA5- and MITA-induced interferon immune responses [52]. 

Among the 20 top up-regulated genes, five different myosin heavy chain encoding genes and a myosin-binding protein h-like were highly expressed, suggesting that myosin plays an important role in D. labrax response to NNV at 14 dpi. Myosins represent a large divergent protein family, with heavy and light chains, and they are involved in muscle and cell contraction, membrane transport, phagocytosis, and motility processes. Μyosin also participate in immune function regulation with a notable role in viral infections [53]. It has been suggested that viral entry involves virus attachment, receptor recruitment, the establishment of a link to the underlying actin cytoskeleton and myosin II–mediated transport of the virus to the cell body prior to cell entry [54]. Different types of myosin have been found to be involved in herpesvirus infection, mediating the entry of the Epstein–Barr virus, participating in capsid assembly, transportation, and release of Kaposi’s sarcoma-associated herpesvirus, facilitating the entry of the Singapore grouper iridovirus into fish cells and the internalization of feline infectious peritonitis virus, and being necessary for efficient production, nuclear egress, and capsid localization during human cytomegalovirus infection. Recently, myosin III light chain b was found to be involved in NNV infection [53]. Overall, it has been proven that myosins facilitate fish virus entry, therefore their up-regulation at 14 dpi might be indicative of a shift in virus entry mechanisms to host cells related to the carrier state, a hypothesis which needs further study by analysis of myosin genes expression levels in more time points post infection.

Transient receptor potential cation channel subfamily m member 2-like (TRPM2) was the most down-regulated gene 14 post NNV infection. Transient receptor potential (TRP) channels are membrane-bound, multi-subunit proteins that aid cation transport (e.g., Ca^2+^, Mg^2+^) across cell membranes. Of all the cations, Ca^2+^ is a universal and versatile carrier of cellular signals and plays a critical role in many cellular functions including cell division, proliferation, migration, transcription, differentiation, cell death, and stress responses. Host cell membranes are embedded with Ca^2+^ receptors, channels, exchangers, and pumps which can be perturbed by viruses facilitating infections in various ways. For example, an increase in cytosolic Ca^2+^ levels activates enzymatic processes and Ca^2+^-dependent transcription factors to promote viral pathogenesis; a decrease in Ca^2+^ levels in ER and Golgi may lead to virus trafficking reduction or inhibit host-mediated anti-viral immune responses, and ER mitochondrial Ca^2+^ regulates apoptosis which can serve as both an anti-viral and a viral-promoting step [55]. TRP channels regulate Ca^2+^ dynamics in host cells; therefore, their expression levels are critical for viral infections. The TRPM channel belongs to the melastatin group of the TRP family, and TRPM2 is a widely distributed, reactive oxygen species-sensitive protein with an important role in the immune system. In fact, HBV infection promotes TRPM2 expression to facilitate virus replication, while TRPM2 inhibition protects cells from H9N2 virus-induced damage. A recent study reported that high-level expression of TRPM2 may facilitate higher replicative ability and pathogenicity of infectious hematopoietic necrosis virus (IHNV) in rainbow trout [56].

The second most down-regulated gene was mitogen-activated protein kinase-binding protein 1-like gene (MAPKBP1). Mitogen-activated protein kinases (MAPKs) are involved in signal transduction pathways with important roles in cytokine gene expression regulation, particularly interleukin-1, and as a key transcription factor in the NF-κB signaling pathway [57]. It has been reported that an extracellular signal-regulated kinase which is involved in MAPK pathway is activated by the white spot syndrome virus (WSSV) in the early stage of *Litopenaeus vannamei* shrimp infection; however, when the kinase is silenced or inhibited, the WSSV proliferation is reduced, and the viral gene transcription is delayed [58]. Therefore, MAPKBP1 down-regulation at 14 dpi may have a regulatory role in virus replication in the virus carrier state. 

In the present work, we followed an approach consisting of two parts to study the effects of NNV infection on *D. labrax* head kidney: a time-course study focused on immune gene expression levels and a total transcriptome analysis on Day 14 post NNV infection. The experimental infection protocol was run up to 28 days, and it was found that on Day 14, the surviving fish retained a high viral load in their brain, while all disease symptoms disappeared. The NNV-infected brains from fish of that time point (14 dpi) were highly infective on SSN-1 cell cultures; therefore, 14 dpi was assumed to be a time point related with the establishment of the virus carrier state. The detailed observation of selected immune responses provided information on the transition from acute to persistent infection, reflected on the gene expression profiles. Since at 14 dpi, there seemed to be a significant time point for the carrier state establishment, a total RNA-seq analysis was performed. It was found that most of the corresponding biological process genes were involved in cellular processes, most of the cellular component genes were corelated with cellular anatomical entity and most of the molecular function genes were classified as binding-related genes. It should be noted that the immune system process as a BP subcategory was only found in up-regulated genes, confirming our results on the detailed immune gene expression levels analysis.

## 4. Materials and Methods

### 4.1. Experimental Fish and NNV Challenge

Healthy, non-vaccinated fish were used throughout the experimentation. All used sea bass were transported by Nireus Aquaculture S.A. at the Laboratory of Ichthyology-Aquaculture and Aquatic Animal Health (ICHTHYAI), Department of Marine Sciences, University of The Aegean, Greece. A total of 500 randomly selected fish (weight: 46.43 ± 13.4 g) were acclimatized for 3–4 days in 1 m^3^ cylindroconical fiberglass tanks connected to a closed recirculated sea water system, with a 20 m^3^ total volume capacity. Water was recirculated via a 14 m^3^ h^−1^ sea water pump, filtered via a sand filter, disinfected via 5 × 39 W UV-C lamps, treated in a biological filter and aerated via air stones connected to five 150 L h^−1^ air pumps. Seawater in the system was renewed by 1/3 every 1.5 months. The seawater temperature was maintained at 22–23 °C during the acclimatization period. Prior to infection, the required number of fish was transported to an allocated tank system where temperature was gradually raised to 27.0–27.2 °C and the fish were acclimatized for 7 more days. The temperature remained constant at 27.0–27.2 °C throughout the experimentation. Salinity was 3.8–3.9‰, dissolved O_2_ was maintained above 4.8 mg L^−1^, total ammonia nitrogen and nitrite were kept below 0.05 ppm and 0.5 ppm, respectively, and nitrate levels were maintained below 40 mg L^−1^. pH ranged between 7.9 and 8.1. Temperature, dissolved oxygen and nitrogen metabolites were measured daily, while salinity and pH were measured on a weekly basis.

The fish were reared in a 12 h light: 1 2h dark photoperiod and were fed with 1–2% of their biomass commercial diet (Feedus, Blueline, Karnataka, India) 3 times a day with 6 h intervals. Each fish carried an individual electronic tag, previously inserted in their abdominal cavity by Nireus Aquaculture S.A. Before the infection experiment, total RNA was extracted from three randomly selected individuals’ brains and amplified with an NNV RT-qPCR assay using the Quantitect Probe RT-PCR master mix (Qiagen, Hilden, Germany) to ensure that those specimens were not infected. The primers and probe which were used are listed in Appendix A.

NNV was originally isolated from naturally infected *D. labrax* fish (genotype: RGNNV [59]), and NNV propagation was performed as previously described [60]. Briefly, fish brains were homogenized in EMEM (Eagle Minimum Essential Medium; Sigma-Aldrich, Steinheim, Germany) or Leibovitz L15 medium (Biochrom, Berlin, Germany) containing 10% FBS (Fetal bovine serum; Biochrom, Berlin, Germany). The homogenates (10% *w*/*v*) were centrifuged (4000× *g*, 15 min, 4 °C) and the resulting supernatant was passed through 0.22 μm filters (Whatman PTFE, 0.22 μm; GE healthcare, Buckinghamshire, England) before inoculation on cell cultures. Following inoculation, the SSN-1 cells (The European Collection of Animal Cell Cultures, Salisbury, UK) were grown at 26 °C in Falcon Primaria cell culture flasks (Becton Dickinson Labware, Franklin Lakes, NJ, USA) containing Leibovitz’s L15 medium, supplemented with 10% FBS, 100 u/mL penicillin, 100 μg/mL streptomycin and 2 mM glutamine (Gibco, Paisley, UK). Virus titration was performed on monolayers of SSN-1 cell grown in a 96-well plate. Viral suspensions were prepared with 12-fold serial dilutions in EMEM supplemented with 10% FBS. Quadruplicates of 50 µL of each dilution were added in a 96-well plate seeded with SSN-1 cells. Cultures were incubated at 26 °C for 6 days. During this period, the cell monolayers were observed for the appearance of the cytopathic effect (CPE), and the final titer, expressed as TCID_50_ mL^−1^, was estimated by the end-point titration method [61].

A summary of the experimental setup is given in Figure 1. On Day 7 after the initial rearing, fish were stocked in groups of 70, in three tanks for each time point included in the study. Sea bass were challenged by intramuscular injection in the dorsal muscle with 7 × 10^6^ TCID_50_ mL^−1^ of a Nodavirus-containing supernatant (100 µL). As a negative control group, uninfected sea bass (mock-challenged with 100 µL PBS) were used, and sampling was performed at the same time points mentioned above. Fish were monitored twice a day and mortalities were recorded. At each specific time point, the deceased fish were removed, and 15 fish of the remaining live population (5 samples from each tank) were randomly selected. Before sampling, the specimens were anesthetized with 0.2% phenoxyethanol and weighed. Blood was drawn by the caudal vein with heparinized syringes and was transferred to 1.5 mL tubes containing 5 mM EDTA. Plasma was isolated by centrifugation at 150× *g* for 10 min at 4 °C. Then, head kidney and brain tissues were removed aseptically and were either subjected in RNA isolation immediately or stored in RNAlater (Qiagen, Hilden, Germany) at −80 °C. The same procedure was followed for 5 fish belonging to the negative control group. For determination of viral load in each time point, a quantitative RT-qPCR assay was used [30] in fish brains (*n* = 5), obtained from each tank. 

### 4.2. Ethics Statement

Fish care and the challenge experiments were conducted according to the guidelines of the institutional committee on Bioethics. Experimentation was performed in the Laboratory of Ichthyology-Aquaculture and Aquatic Animal Health (ICHTHYAI) (Government Issue 1255/28-4-2016). ICHTHYAI was granted all the required permits for producing (EL 83 BIObr 01), supplying (EL 83 BIOsup 01) and experimenting on aquatic organisms (EL83 BioExp 01), according to the Presidential Decree 56/2013 conforming to Directive 2010/63/ΕΕ (Decision No 4053/14-3-2017 of the competent Regional Veterinary Authority). Fish were euthanized using a procedure listed on the appropriate license, and the protocol for the experimental infection performed in this study was approved by Decision No 5379/4-4-2017 of the competent Regional Veterinary Authority. 

### 4.3. Biometrical Data

Body, spleen, head kidney and brain weights were measured for each fish (*n* = 15 challenged fish and *n* = 5 non-challenged fish per time point), as well as the fish total length (TL, from snout to edge of caudal fin). The growth performance was assessed using the weight gain (WG % = [(final weight − initial weight) × initial weight^−1^] × 100), the specific growth rate (SGR % = [(ln final weight − ln initial weight) × number days^−1^] × 100) and the condition factor (CF % = [(weight × length^−3^)] × 100) calculated values [62]. The splenosomatic (SSI) index was calculated as the ratio between organ weight and body weight, i.e., SSI% = [(spleen weight (mg) × body weight (mg)] × 100 [63].

### 4.4. Haematological Parameters 

Hematocrit (Hct) was measured by the microhematocrit method [64]. Briefly, blood was adsorbed by capillary action in hematocrit measurement tubes. The tubes were centrifuged at 10,700 rpm for 3 min in hematocrit centrifuge (Hettich–Haematokrit 210, Tuttlingen, Germany). The volumetric content of the sedimented erythrocytes were read off a scale as the percentage of the total blood volume (% Hct). Leukocrit (% Lct) was evaluated, respectively.

### 4.5. Measurement of Specific Anti-NNV and Total Sea Bass Serum Antibodies

#### 4.5.1. Rabbit Anti-NNV Polyclonal Antibodies and Hyperimmune Sea Bass Anti-NNV Serum Preparation

Anti-NNV polyclonal antibodies were prepared as previously described [60]. Serum was titrated against the NNV antigen using a simple indirect ELISA and stored at −20 °C. Healthy unvaccinated sea bass (average weight: 34 g) were injected intraperitoneally (i.p.) with a homogenized mixture containing 100 μL of b-nodavirus solution (100 μg of inactivated (70 °C, 2 h) b-nodavirus protein per mL) and 100 μL of complete Freund’s adjuvant (CFA) (Sigma-Aldrich; Steinhem, Germany) or incomplete Freund’s adjuvant (IFA) (Sigma-Aldrich; Steinhem, Germany) antigen (i.e., 1:1 mixture). Fish were injected at Days 0, 28 and 45. On Day 50, the fish were anesthetized with 2% phenoxyethanol; blood was collected from the caudal vein and allowed to clot overnight at 4 °C, centrifuged at 150× *g* for 10 min to separate the serum, and stored at −20 °C. Serum samples were titrated against the b-nodavirus antigen as follows: plates were coated with the b-nodavirus antigen and blocked, and serial dilutions of the hyperimmune serum were added. Subsequently, an anti-sea bass IgM monoclonal antibody (product FO1, Aquatic diagnostics Ltd., Scotland, UK) was added, followed by the addition of anti-mouse IgG-biotin (Sigma-Aldrich; Steinhem, Germany) and extravidin-HRP (Sigma-Aldrich; Steinhem, Germany). The optimum dilution result (1:8) compared to negative control was selected for use as positive control in further studies [65]. 

#### 4.5.2. Determination of Specific Anti-NNV Antibody Amount

A modified sandwich ELISA was utilized for specific anti-NNV IgM level determination [66]. Briefly, 96-well plates were coated with rabbit anti-NNV serum (1:5000), followed by the addition of a NNV suspension (10^8.2^ TCID_50_ mL^−1^). The serum samples were added (diluted 1:8 in 1% BSA, 0.1% Tween 20, 0.02 M PBS, pH 7.2), and incubation with anti-sea bass IgM monoclonal antibodies was performed. Subsequently, anti-mouse IgG biotin conjugate (1:1750) was added, followed by extravidin-HRP (1:500) addition. Results were visualized using tetramethyl–benzidine–HCl (TMB) (Sigma-Aldrich), and reaction was stopped with the addition of 2M H_2_SO_4_. Results were read on a MR-96A microplate reader (MINDRAY, Shenzen Mindray Bio-medical Electronics Co., Ltd., Shenzhen, China) at 450 nm. 

To determine the cut-off limit for antibodies from negative control samples, the mean value (*n* = 5) and the standard deviation of the resulting absorptions were calculated. The cut-off value was defined as the average of the absorptions plus the standard deviation multiplied by 3. Therefore, a sample had a positive response when its signal was above the cut-off value.

### 4.6. RNA Isolation and Complementary DNA (cDNA) Synthesis

For each fish (*n* = 9 challenged fish and *n* = 5 non-challenged fish per time point), total RNA from the head kidney and the brain was extracted, as follows: tissue from the samples at each time-point was homogenized using the TissueLyzer mechanical homogenizer (Qiagen, Hilden, Germany) with 5 mm steel beads (Qiagen), and the RNeasy Mini Kit (Qiagen) was used for total RNA extraction, following the manufacturer’s instructions. RNA quantity and quality were assessed by spectrophotometry (NanoDrop 2000; Thermo Fisher Scientific, Wilmington, DE, USA). Only RNA samples of high quality were used for constructing the cDNA libraries. Total RNA was used as a template to synthesize cDNA using QuantiNova Reverse Transcription kit (Qiagen) according to the manufacturer’s instructions. Approximately 5 μg RNA were used as input material.

### 4.7. Gene Expression Analysis

Real-time PCR assays were carried out to analyze the expression pattern of different immune-relevant genes in European sea bass fish challenged with nodavirus. Samples were taken from the nodavirus-challenged (*n* = 9) and non-challenged (mock/non-infected) (*n* = 5) fish at 0, 3, 6, 12, 24 and 48 hpi, followed by 3, 4, 7, 14 and 28 dpi. Specific primers used for gene expression analysis (Appendix A) were designed utilizing Primer-Blast (http://www.ncbi.nlm.nih.gov/tools/primer-blast/ last accessed on 15 June 2022), IDT_Primer Quest (https://eu.idtdna.com/PrimerQuest/Home/Results/ last accessed on 15 June 2022) and Primer3Plus (http://www.bioinformatics.nl/cgi-bin/primer3plus/primer3plus.cgi/ last accessed on 15 June 2022). The potential primer secondary structures (homo- or cross-dimers, hairpin structures) and primer specificity were checked with IDT_Primer Quest and Primer-Blast, respectively. All qRT-PCR primers were designed according to the minimum information needed for publication of qRT-PCR experiments (MIQE) guidelines [67]. Five-point standard curves of 4-fold dilution series (1:1–1:256) from pooled cDNA were used for calculation of the PCR efficiency, given by equation E% = (10^1/slope^ − 1) × 100. The slope was calculated from the linear regression model fitted from the log-transformed cDNA concentrations plotted against the Ct values. The PCR efficiencies for each primer pair are listed in Appendix A. PCR efficiencies between 90 and 110% were considered acceptable. The b-actin gene was chosen in our study as a reference gene [37]. Before sample quantification experiments, the specificity of each primer pair was studied using positive and negative samples. 

Real-time PCR reactions were performed with QuantiNova SYBR Green PCR (Qiagen) using 1 µL of a 1:10 dilution of cDNA. Primers for all genes were used at 500 nM. An RNA sample (prepared as previously described), a cDNA synthesis master mix, and ddH_2_O were used as real-time PCR internal amplification controls. The thermal conditions used were as follows: 2 min at 95 °C of pre-incubation followed by 40 cycles at 95 °C for 10 s and 60 °C for 30 s. An additional temperature ramping step was utilized to produce melting curves from 62 to 95 °C to verify the amplification of a unique single product on all samples. All reactions were performed in technical triplicate using a RotorGene Q PCR Detection System (Qiagen). 

Quantification was performed according to the comparative C_T_ method [68]. The value for each experimental condition was expressed as a normalized relative expression, calculated in relation to the values of control group and normalized against those of the reference gene (by its geometric average). The results were expressed as the average of values obtained at 0, 3, 6, 12, 24 and 48 hpi; and 3, 4, 7, 14 and 28 dpi (*n* = 3 fish per tank per time point). A melting curve analysis of the amplified products validated the primers for specificity.

### 4.8. NGS Sample Preparation

To perform total RNA sequencing, total RNA from the head kidney was extracted using a Trizol-based protocol. Tissues (~75 mg) from NNV-infected (*n* = 5) and non-challenged (*n* = 4) fish at a 14 dpi time point were homogenized using the TissueLyzer mechanical homogenizer with 5 mm steel beads (Qiagen) as described above. The TRIzol^®^ Reagent solution (Invitrogen, Carlsbad, CA, USA) was used for total RNA extraction, following the manufacturer’s instructions. RNA quantity and quality were assessed by spectrophotometry (NanoDrop 2000) and a Qubit 2.0 fluorometer (Thermo Fisher Scientific, Waltham, MA, USA), and the RNA integrity number (RIN) was measured using a 2100 Bioanalyzer instrument (Agilent Technologies, Santa Clara, CA, USA). Only RNA samples of high quality (RIN ≥ 7.5) were used for constructing the cDNA libraries, and each samples’ transcriptome was analyzed individually. 

### 4.9. cDNA Library Construction, Sequencing and Transcriptome Mapping

RNA samples were used to create the sequencing libraries by using the Ion Total RNA-Seq Kit v2 kit (Thermo Fisher Scientific, Waltham, MA, USA). The 9 libraries were prepared according to the manufacturer’s instructions and then sequenced using Ion Torrent technology (Ion S5XL). Raw data were processed to remove adapters and were normalized for inherent systematic or experimental biases, using the Bioconductor package DESeq2. The reads were aligned to the *Dicentrarchus labrax* (seabass_V1.0—GCA_000689215.1) reference genome [10] with hisat2 and bowtie2. Post-mapping quality control and basic differential expression analysis was performed with the Bioconductor package metaseqR2 [69,70]. A transcript was considered as a differentially expressed transcript if the *p*-value threshold was less than 0.05 (significance level) and the log2-transformed fold change (log2FC) was more than 1.

### 4.10. Annotation of Differentially Expressed Genes, Functional Analysis and Construction of a Protein–Protein Interaction Network

The function of the identified differentially expressed transcripts was analyzed in OmicsBox software (Version 2.2.4) by first using BLASTX against an NCBI non-redundant (NR) database to search for the possible top hit proteins (accessed on 8 December 2022). Thereafter, blasted sequences were subjected to gene ontology (GO) mapping and annotation with default parameters. Subsequently, the differentially expressed transcripts were manually searched for possible protein orthologs of zebrafish (Taxid: 7955) in the Uniprot database (accessed on 27 January 2023). The Search Tool for the Retrieval of Interacting Genes/Proteins database (STRING v11.5) was subsequently used to construct their PPI network. Given a list of the proteins as input, STRING found their neighboring interactors and generated the PPI network consisting of all these proteins and all the interactions between them. All the interactions between them were derived from high-throughput lab experiments and previous knowledge in curated databases at a high level of confidence (sources: experiments, databases; score ≥ 0.90) [71].

## 5. Conclusions

Despite the complex cell-intrinsic and intercellular antivirus mechanisms available at the host’s disposal, viruses have developed various means that counteract these mechanisms and allow for virus persistence [72]. Taken together, our results suggest that the establishment of an NNV virus carrier state involves increased immune gene expression in the head kidney, especially for interferon pathway-related genes and cytokines, as seen in the first part of our analysis, and a more “mechanistic” attempt of the host to control virus infection by de-regulation of regulatory (TRIM16L, TRPM2, MAPKBP1) and myosin genes, as observed on the transcriptome analysis. The present study provides an insight into the efforts of the host cells to ovecome the infection by observing how the immune system operates dynamically in living, virally infected tissues. However, further studies are needed to elucidate the underlying mechanisms of NNV persistance when the host is in the carrier state. Moreover, the assessment of the protein levels associated with the present study will be beneficial for upgrading the present knowledge for NNV infection effects on teleosts.

## Figures and Tables

**Figure 1 ijms-24-16613-f001:**
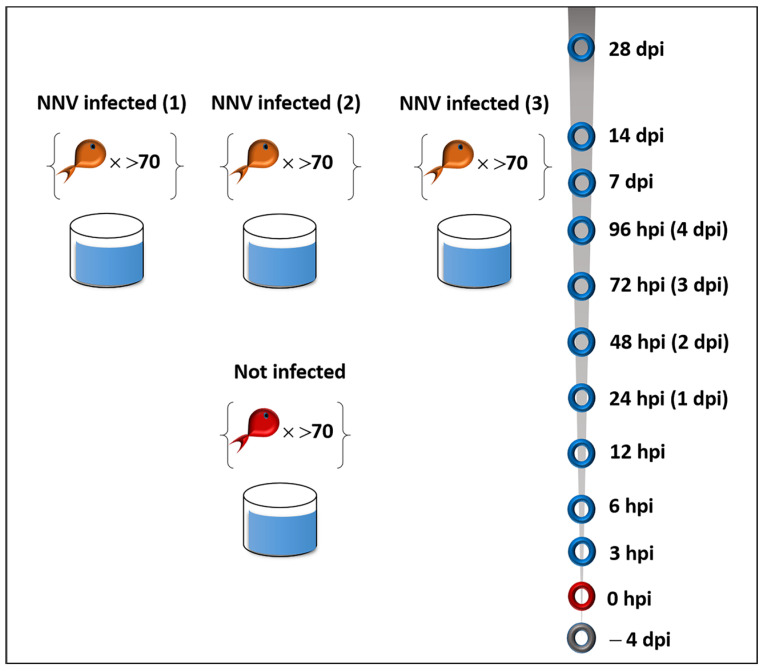
Experimental setup (hpi: hours post infection, dpi: days post infection).

**Figure 2 ijms-24-16613-f002:**
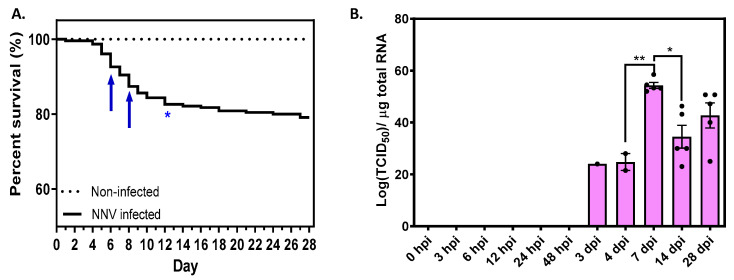
(**A**) Cumulative survival of NNV-challenged (continuous line) and non-infected (dashed line) fish groups. The mean mortality is represented in each time point (days post infection). The arrows indicate the time points with the highest daily mortalities. One asterisk indicates *p* < 0.1. (**B**) Absolute RNA1 quantification in the brain from European sea bass challenged with NNV. Results are mean ± SEM (*n* = 5). Each dot represents a biological sample. Significant differences were analyzed using a one-way ANOVA test with Bonferroni’s multiple comparison test correction. One asterisk indicates *p* < 0.05, two asterisks indicate *p* < 0.01.

**Figure 3 ijms-24-16613-f003:**
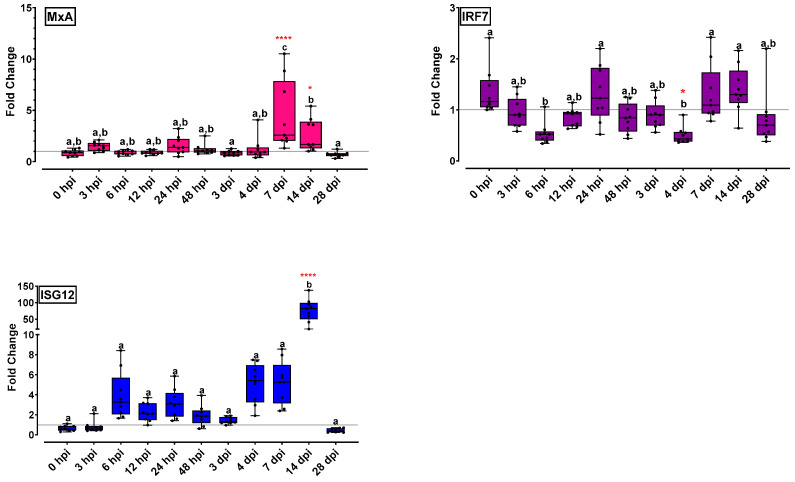
Interferon pathway-related gene expression (MxA, IRF7, ISG12) levels for the head kidney of NNV-challenged and non-challenged groups are given as bounds of box and whisker plots with min-to-max values. The line represents the median (*n* = 9 fish per group), the whiskers show the data range, and the box shows the interquartile range. Each dot represents a biological sample. All statistical differences were assessed with ordinary two-way ANOVA followed by Tukey’s multiple comparison test. Statistically important differences between challenged and non-challenged groups are denoted with red asterisks (*: *p* < 0.05, ****: *p* < 0.0001). Different lower-case letters denote statistically significant differences between sampling time points for the NNV challenged group.

**Figure 4 ijms-24-16613-f004:**
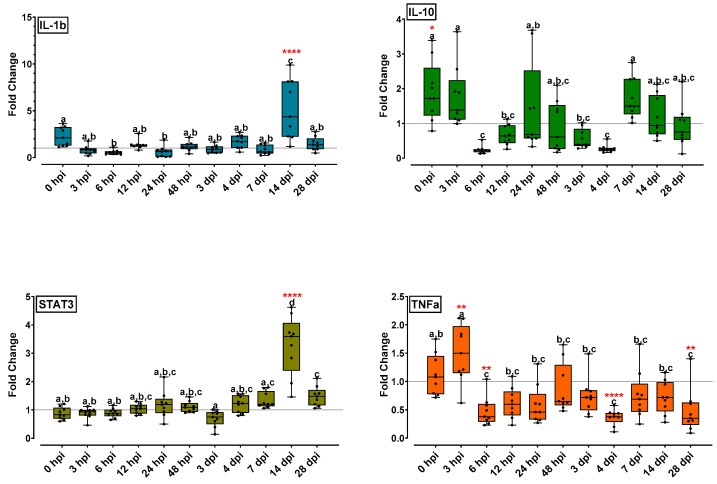
Cytokine-related gene expression (IL-1b, IL-10, STAT3, TNFa) levels for the head kidney of NNV-challenged and non-challenged groups are given as bounds of box and whisker plots with min-to-max values. The line represents the median (*n* = 9 fish per group), the whiskers show the data range, and the box shows the interquartile range. Each dot represents a biological sample. All statistical differences were assessed with ordinary two-way ANOVA followed by Tukey’s multiple comparison test. Statistically important differences between challenged and non-challenged groups are denoted with red asterisks (*: *p* < 0.05, **: *p* < 0.01, **** indicate *p* < 0.0001). Different lower-case letters denote statistically significant differences between sampling time points for the NNV challenged group.

**Figure 5 ijms-24-16613-f005:**
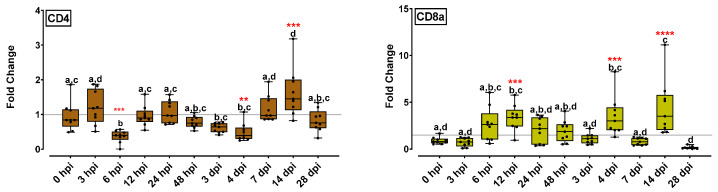
T-cell marker-related gene expression (CD4, CD8a) levels for the head kidney of NNV-challenged and non-challenged groups are given as bounds of box and whisker plots with min-to-max values. The line represents the median (*n* = 9 fish per group), the whiskers show the data range, and the box shows the interquartile range. Each dot represents a biological sample. All statistical differences were assessed with ordinary two-way ANOVA followed by Tukey’s multiple comparison test. Statistically important differences between challenged and non-challenged groups are denoted with red asterisks (**: *p* < 0.01, ***: *p* < 0.001, ****: *p* < 0.0001). Different lower-case letters denote statistically significant differences between sampling time points for the NNV = challenged group.

**Figure 6 ijms-24-16613-f006:**
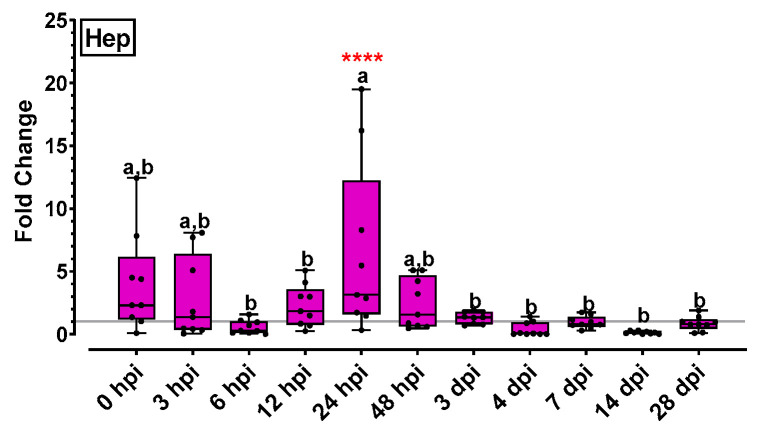
Antiviral peptide gene expression (hepcidin) levels for the head kidney of NNV-challenged and non-challenged groups are given as bounds of box and whisker plots with min-to max values. The line represents the median (*n* = 9 fish per group), the whiskers show the data range, and the box shows the interquartile range. Each dot represents a biological sample. All statistical differences were assessed with ordinary two-way ANOVA followed by Tukey’s multiple comparison test. Statistically important differences between challenged and non-challenged groups are denoted with red asterisks (****: *p* < 0.0001). Different lower-case letters denote statistically significant differences between sampling time points for the NNV-challenged group.

**Figure 7 ijms-24-16613-f007:**
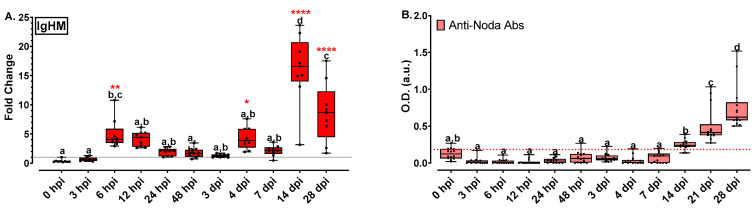
(**A**) Immunoglobulin gene expression (IgHM) levels for the head kidney of NNV-challenged and non-challenged groups are given as bounds of box and whisker plots with min-to-max values. The line represents the median (*n* = 9 fish per group), the whiskers show the data range, and the box shows the interquartile range. Each dot represents a biological sample. (**B**) Specific anti-nodavirus antibody O.D. levels for the NNV-challenged group are given as bounds of box and whisker plots with min-to-max values. The line represents the median (*n* = 15 fish per group), the whiskers show the data range, and the box shows the interquartile range. All statistical differences were assessed with ordinary two-way ANOVA followed by Tukey’s multiple comparison test. Statistically important differences between challenged and non-challenged groups are denoted with red asterisks (*: *p* < 0.05, **: *p* < 0.05, ****: *p* < 0.0001). Different lower-case letters denote statistically significant differences between sampling time points for the NNV challenged group.

**Figure 8 ijms-24-16613-f008:**
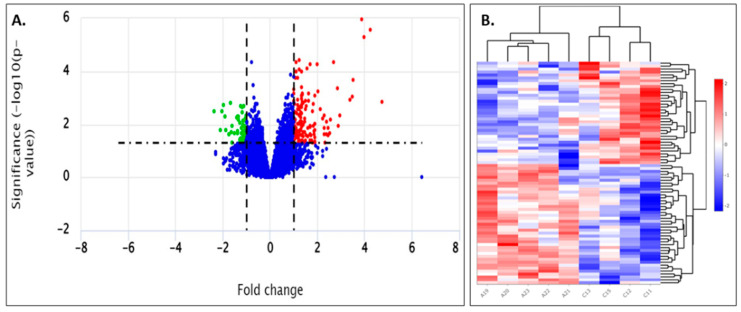
(**A**) Volcano plot of differentially expressed genes in *D. labrax* head kidney 14 days post infection with NNV. Red dots and green dots represent the up-regulated and down-regulated differentially expressed genes, respectively. (**B**) Cluster analysis of DEGs. Each column in the graph represents a sample, each row represents a gene, and the expression of genes in different samples is represented by different colors, with redder colors indicating higher expression and bluer colors indicating lower expression. A19–A23: NNV-infected *D. labrax*; C11–C15: non-infected samples.

**Figure 9 ijms-24-16613-f009:**
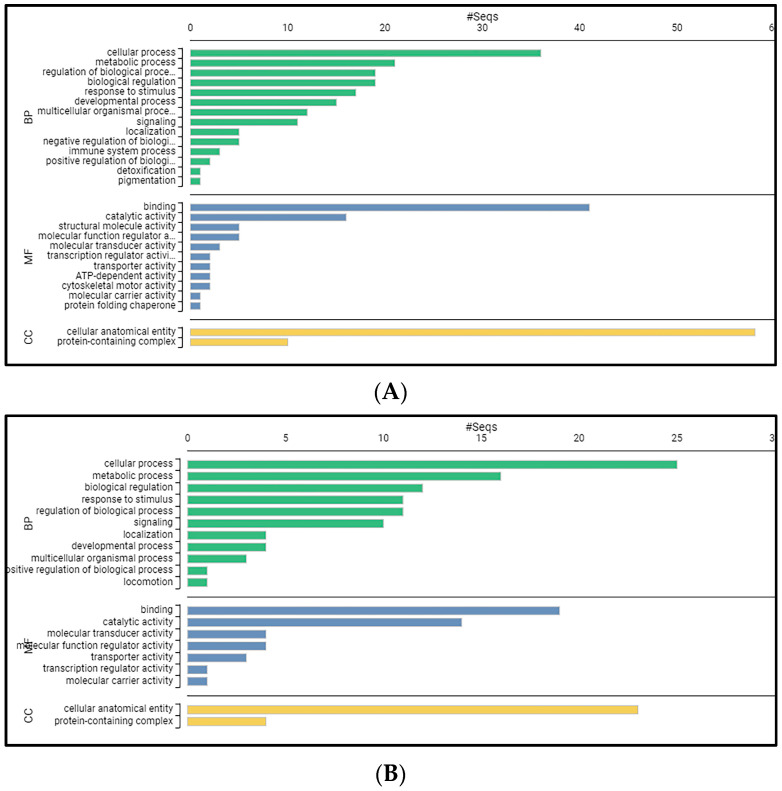
Gene ontology (GO) annotation of the differentially expressed genes in the three main GO categories: biological process, cellular component, and molecular function in *D. Labrax* head kidney 14 days post infection with NNV. (**A**) Up-regulated genes. (**B**) Down-regulated genes. BP: Biological process, MF: molecular function, CC: cellular component.

**Figure 10 ijms-24-16613-f010:**
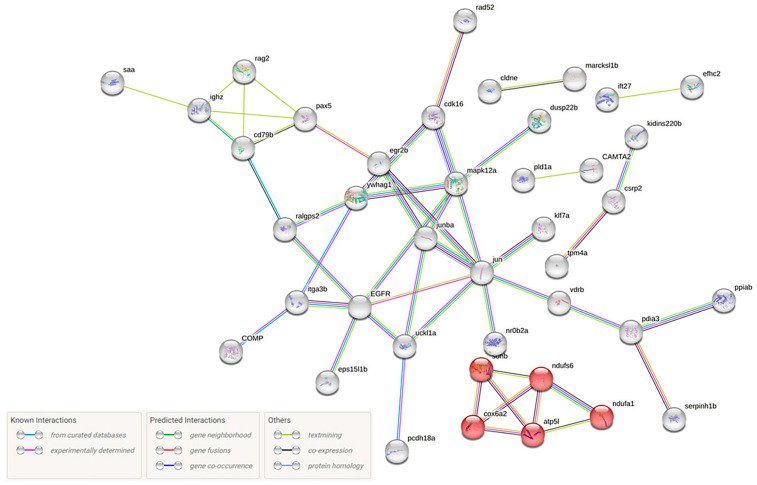
Protein–protein interaction (PPI) networks for the differentially expressed proteins found 14 days following NNV infection in *D. labrax* head kidney. This was retrieved via API access to the STRING database (https://string-db.org) (accessed on 23 February 2023) and was performed based on the *Danio rerio* protein database. Each node represents a differentially expressed protein. Red nodes represent the significantly enriched KEGG pathway oxidative phosphorylation. The edges represent protein–protein interactions, and the nature of the protein–protein interactions are color-coded, as indicated in the figure.

**Table 1 ijms-24-16613-t001:** Top 20 up-regulated NNV-challenged *D. labrax* genes 14 days post infection.

Gene ID	Description	Gene Symbol	Log2FC	*p*-Value
DLAgn_00266700	tripartite motif-containing protein 16-like	-	4.7558	0.0013
DLAgn_00246320	myosin heavy chain	MYHZ1.1	3.8984	1.08 × 10^8^
DLAgn_00231310	uncharacterized protein loc101466409 isoform x1	-	3.4739	0.0009
DLAgn_00244650	myosin heavy chain	MMYHL1	3.3833	0.0012
DLAgn_00216380	myosin heavy chain	MYHZ1.1	2.9484	0.0048
DLAgn_00096300	myosin-binding protein h-like	MYBPHA	2.5482	0.0069
DLAgn_00252300	ubiquitin carboxyl-terminal hydrolase 35	-	2.5146	0.0209
DLAgn_00118970	kallikrein-8	KLK8	2.4557	0.0139
DLAgn_00266850	proteinase-activated receptor 4	F2RL3	2.4476	0.0256
DLAgn_00232470	uncharacterized protein iii	-	2.4018	0.0231
DLAgn_00159040	serum amyloid a	SAA	2.3315	0.0479
DLAgn_00234950	myosin heavy chain	MMYHL1	2.3037	0.0134
DLAgn_00206120	cysteine and glycine-rich protein 2	CSRP2	2.3026	0.0231
DLAgn_00080910	peptidyl-prolyl cis-trans isomerase-like	PPIAB	2.2970	0.0046
DLAgn_00045390	solute carrier family 22 member 5-like	SLC22A5	2.2270	0.0062
DLAgn_00251790	myosin heavy chain	MYHZ1.1	2.1442	0.0026
DLAgn_00247100	endonuclease domain-containing 1	-	1.9130	0.0097
DLAgn_00062850	tetranectin-like	TETN	1.9005	0.0451
DLAgn_00167280	leptin	LEP	1.9005	0.0480
DLAgn_00266930	histone-lysine n-methyltransferase mll3	MLL3A	1.8973	0.0008

**Table 2 ijms-24-16613-t002:** Top 20 down-regulated NNV-challenged *D. labrax* genes 14 days post infection.

**Gene ID**	**Description**	**Gene Symbol**	**Log2FC**	***p*-Value**
DLAgn_00049910	transient receptor potential cation channel subfamily m member 2-like	TRPM2	−2.3655	0.0033
DLAgn_00227410	mitogen-activated protein kinase-binding protein 1-like	MAPKBP1	−2.1043	0.0159
DLAgn_00218080	kinase d-interacting substrate of 220 kda-like	KIDINS220B	−2.0489	0.0033
DLAgn_00251680	butyrophilin subfamily 1 member a1-like	MOG	−1.9534	0.0018
DLAgn_00221860	cytochrome p450 2j2-like	CYP2AD2	−1.8864	0.0052
DLAgn_00018460	calmin delta	CLMN	−1.8694	0.0159
DLAgn_00249930	zinc finger protein 850-like	-	−1.6659	0.0016
DLAgn_00231790	hippocampus abundant transcript 1	HIAT1	−1.6471	0.0191
DLAgn_00099340	hemoglobin beta chain	-	−1.6142	0.0469
DLAgn_00063590	cub and sushi domain-containing protein 3-like	CSMD3	−1.6049	0.0113
DLAgn_00130980	solute carrier family 26 member 10	SLC26A10	−1.4112	0.0057
DLAgn_00142520	sodium calcium exchanger 1-like	SLC8A4A	−1.3812	0.0221
DLAgn_00241570	sterile alpha motif domain-containing protein 9-like	SAMD9L	−1.3797	0.0175
DLAgn_00218000	tgf-beta receptor type-1-like	TGFBR1	−1.3389	0.0068
DLAgn_00151800	ral gef with ph domain and sh3 binding motif 2	RALGPS2	−1.3248	0.0051
DLAgn_00017450	piezo-type mechanosensitive ion channel component 1	PIEZO1	−1.2886	0.0361
DLAgn_00203250	sperm acrosome membrane-associated protein 4	SPACA4	−1.2872	0.0402
DLAgn_00252660	general transcription factor ii-i repeat domain-containing protein 2-like	-	−1.2695	0.0088
DLAgn_00134630	plexin-partial	PLXNB1	−1.2363	0.0020
DLAgn_00227470	v-set domain-containing t-cell activation inhibitor 1-like	-	−1.2264	0.0325

## Data Availability

The raw sequencing data are available in the NCBI database (accession no. PRJNA1030357).

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
