# Peer review of "Nervous Necrosis Virus Modulation of European Sea Bass (Dicentrarchus labrax, L.) Immune Genes and Transcriptome towards Establishment of Virus Carrier State"

_ijms, 2023, doi:10.3390/ijms242316613_

Round 1

Reviewer 1 Report (New Reviewer)

Comments and Suggestions for Authors

Authors of present study have evaluated changes in gene expression of D. labrax head kidney upon infection with nervous necrosis virus over the period of 28 days post viral infection. Authors have carefully analyzed gene expression patterns of cytokines, immunoglobulins, T-cell markers, interferon pathways, antiviral peptides, etc. Going further, the authors have also performed transcriptome analysis of differentially expressed genes at 14 dpi which seems to be the time point having most significant changes in various gene expression levels. 

A few minor comments can be addressed which are as follows:

  1. A general suggestion is to improve section headings so as to give more insight into results described in the respective sections rather than only listing them with names of pathways.

  1. Fig2A: How does the variance look for % survival between 3 infection groups? Can authors provide survival curves for 3 separate infection groups?

  1. Can authors explain in detail what condition factor parameters (line 159) and splenosomatic index (161) are and what they signify? How was the specific growth rate calculated?

  1. FigS1 and S2: It is not clear what the p-values are for the statistical significance.

  1. Fig3: Lower case letters denote statistically significant differences between timepoints as mentioned in figure legend; however it is not clear if they are being compared to 0hpi or something else? Can authors clarify this? IRF7 levels in Fig3 for example- authors say in the text that they are increased on day 7 post infection however, the level seems to be similar to 0hpi. Can authors clarify how they are evaluating these differences?

  1. Fig3: It is clear that levels of MxA and ISG12 tend to increase after infection upto 14 dpi. However, levels of IRF7 seem to fluctuate. Can authors comment more about this?

  1. Fig9 and Fig10 quality should be improved.

Author Response

Response to Reviewer 1 Comments

We would like to thank the time and effort the reviewer 1 put on the comments of our manuscript. We believe that the corrections/additions based on the reviewers’ comments have greatly improved the quality of our manuscript. Here you will find all original comments and the updated information that was included to clarify our results.

Point 1: A general suggestion is to improve section headings so as to give more insight into results described in the respective sections rather than only listing them with names of pathways.

Response 1: All results subheadings have been improved according to reviewer 1 suggestion, to contain a synopsis of the respective section experimental results.

Point 2: Fig2A: How does the variance look for % survival between 3 infection groups? Can authors provide survival curves for 3 separate infection groups?

Response 2: The following phrase has been added in section 2.1 of the manuscript “The cumulative survival rates in each experimental tank with NNV infected fish are also presented in Figure S1, where it is confirmed the absence of significant variations in fish mortality between tanks”. The survival curves for the 3 infection groups (tanks) are provided in supplementary material as figure S1.

Point 3: Can authors explain in detail what condition factor parameters (line 159) and splenosomatic index (161) are and what they signify? How was the specific growth rate calculated?

Response 3: The condition factor (K) of a fish reflects physical and biological circumstances and fluctuations by interaction among feeding conditions, parasitic infections and physiological factors. This also indicates the changes in food reserves and therefore an indicator of the general fish condition [Datta SN, Kaur VI, Dhawan A, Jassal G. Estimation of length-weight relationship and condition factor of spotted snakehead Channa punctata (Bloch) under different feeding regimes. Springerplus. 2013 Sep 4;2:436. doi: 10.1186/2193-1801-2-436].

Splenosomatic index is the weight of the spleen expressed as a percentage of total body weight. Alterations in this index could indicate an abnormal condition in the spleen such as necrosis or swelling due to infection [Gupta N, Gupta DK, Sharma PK. Condition factor and organosomatic indices of parasitized Rattus rattus as indicators of host health. J Parasit Dis. 2017 Mar;41(1):21-28. doi: 10.1007/s12639-015-0744-3].

Both parameters are widely used to estimate fish health, especially in fish feed oriented studies. The above mentioned explanations have been added as supplementary material.

As described in materials and methods section 4.3 the specific growth rate SGR is given by the formula: % = [(ln final weight – ln initial weight)´number days−1] x 100).

Point 4: FigS1 and S2: It is not clear what the p-values are for the statistical significance.

Response 4: In figures S1 – S2, statistical differences were assessed with two-way ANOVA followed by Tukey’s multiple comparison test, as stated in their respective legends. The statistically significant differences were depicted using the Compact letter display statistical method instead of using the asterisks (or p-value) method, because the later resulted in very complicated figures. The Compact Letter Display (CLD) is a statistical method to clarify the output of multiple hypothesis testing when using the ANOVA and Tukey's range tests. CLD facilitates the identification of variables, or factors, that have statistically different means (or averages) vs. the ones that do not have statistically different means (or averages). The basic technique of compact letter display is to label variables by one or more letters, so that variables are statistically indistinguishable if and only if they share at least one letter. Each variable that shares a mean that is not statistically different from another one will share the same letter (Wikipedia). Therefore, the use of significance levels (e.g. p-values) it’s not applicable in the selected depiction. However, for readers’ facilitation, the p-values have been added in figures S1 and S2 (S2 and S3 in the revised manuscript).

Point 5: Fig3: Lower case letters denote statistically significant differences between timepoints as mentioned in figure legend; however it is not clear if they are being compared to 0hpi or something else? Can authors clarify this? IRF7 levels in Fig3 for example- authors say in the text that they are increased on day 7 post infection however, the level seems to be similar to 0hpi. Can authors clarify how they are evaluating these differences?

Response 5: As described in results section 2.5 ‘B-actin was employed as reference gene and the gene expression of the non-infected group at each time point was used to determine the fold change for each gene in the infected group’. The fold change calculation methodology is also described in the materials and methods section 4.7: ‘The value for each experimental condition was expressed as normalized relative expression, calculated in relation to values of control group (i.e. non challenged fish) and normalized against those of the reference gene (i.e. b-actin) (by its geometric average)’. Therefore, gene expression in each time point is evaluated in comparison to the non-infected respective group and the differences described in the text are referring to adjacent time points unless it is otherwise stated.

Point 6: Fig3: It is clear that levels of MxA and ISG12 tend to increase after infection upto 14 dpi. However, levels of IRF7 seem to fluctuate. Can authors comment more about this?

Response 6: As described in the discussion section of the manuscript ‘Members of the IRF family seem to be important in the immune response of neurological tissues of higher vertebrates to RNA viruses [29], and in teleosts, IRF7 has been identified as a primary transcription factor priming the type I IFN response”. NNV is a neurotropic virus, therefore we would expect IRF7 up-regulation by infection, analogously to higher vertebrates. On the other hand, head kidney is considered the fish main immune organ however is not a neurological tissue. As mentioned, ‘a study on IRF7 modulation by viral infections in Asian seabass have demonstrated that high transcript abundance was observed 24 h post infection in brain, heart, kidney and spleen [43] and irf7 was highly expressed 3 dpi in sea bass correlated with the transcription of mxA and isg12 [3]’. The present study is analysing many early- and later time points of the infection, in order to shed light in IRF7 detailed expression profiles as a first step to analyse these fluctuations. More studies on IRF7 gene expression combining head kidney and other tissues (e.g. brain) are in our future plans to gain more understanding of IRF7 role in NNV infection.

Point 7: Fig9 and Fig10 quality should be improved.

Response 7: Figures 9 and 10 qualities were further improved.

Reviewer 2 Report (New Reviewer)

Comments and Suggestions for Authors

The study by Toubanaki presented a study on transcriptome analysis in Nervous Necrosis virus infection in European sea bass. The results indicated a significant change in the expressions of various  immune genes. The overall design of this study is sound and most results are supportive to establish the conclusions. This review has some minor points for authors to improve their paper.

1) The sample collection (batch) for each group is >70. Please mention if the study takes into account individual differences and discuss this as a major limitation.

2) Any experimental results from protein level, either WB or 2D gel, for Figs. 3-7? Also discuss more on the future direction hinted from this paper's results.

3) Incease the font size of all figures, many of which are difficult to read.

4) As mentioned, please add a section to discuss the limitations and future work in discussion section.

Comments on the Quality of English Language

Further proofreading is required.

Author Response

Response to Reviewer 2 Comments

We would like to thank reviewer 2 for the time and effort put on the comments for our manuscript. We believe that the corrections/additions based on the reviewers’ comments have greatly improved the quality of our manuscript. Here you will find all original comments and the updated information that was included to clarify our results.

Point 1: The sample collection (batch) for each group is >70. Please mention if the study takes into account individual differences and discuss this as a major limitation.

Response 1: As mentioned in methods section 4.1 and results section 2.5 ‘Fish were stocked in groups of 70, in three tanks for each time point included in the study. … . Of the 15 fish sampled per time point for infected group, 9 fish were employed on gene expression studies, namely those who had adequate total RNA with the best quality for cDNA preparation’. The described experimental design was carefully chosen in order to take into account fish individual differences as well as possible differences between the experimental tanks. Therefore, the fish which were finally analysed were considered representative of the studied parameters. Furthermore, in order to include the fish individual differences, we chose the box and whisker plots with min-to-max values and all individual values depiction for the gene expression studies. Overall, the individual differences of biological samples are a parameter which cannot be avoided in similar studies, therefore we analysed a representative number of samples and we discuss for trends in gene expression.

Point 2: Any experimental results from protein level, either WB or 2D gel, for Figs. 3-7? Also discuss more on the future direction hinted from this paper's results.

Response 2: Scope of the present study was the assessment of immune genes expression and transcriptome analysis of the NNV infection, as described in the manuscript title. Therefore, we were focused in the molecular level and no WB or 2D experiments were performed. We are also interested to look into protein expression levels and we are planning to run proteomic analysis in the analyzed samples, which is not feasible at the moment due to cost constraint. For that reason, in the discussion section is stated that ‘the assessment of the respective protein levels in a future study will be beneficial for immune responses understanding in depth’.

Point 3: Incease the font size of all figures, many of which are difficult to read.

Response 3: Fonts in figures were increased as suggested.

Point 4: As mentioned, please add a section to discuss the limitations and future work in discussion section.

Response 4: The above discussed comments were incorporated in the manuscript as suggested.

This manuscript is a resubmission of an earlier submission. The following is a list of the peer review reports and author responses from that submission.

Round 1

Reviewer 1 Report

Comments and Suggestions for Authors

The study describes the course of the Nervous necrosis virus (NNV) infection in Dicentrarchus labrax (European sea bass) and the persistence of the virus in survivors. The study focused on the gene expression analysis of selected immune-related genes and transcriptome profiling of the head kidney of D. labrax 14 days post-infection (dpi). The findings indicate that immune-related genes showed differential expression, suggesting their involvement in both early antiviral responses and disease persistence. The differential expression of genes in the head kidney of asymptomatic carriers with a high viral load was also studied. Overall, the study provides insights into the persistent profile of NNV in D. labrax and the host's attempts to control the infection in asymptomatic carriers.

Major comments:

It would help if the authors could elaborate more on these questions below:

Methods:

1)    Can you provide more details about the specific criteria used to design the primers for gene expression analysis? How were the potential secondary structures (homo- or cross-dimers, hairpin structures) and primer specificity checked?

2)    Could you elaborate on how the minimum information needed for publication of qRT-PCR experiments (MIQE) guidelines were followed?

3)    How was the reference gene, b-actin, chosen for this study? Were there any validation experiments conducted to confirm its stability under the experimental conditions?

4)    When performing BLASTX against the NCBI non-redundant (NR) database, what was the e-value threshold used to determine the significance of the hits? Were there any filters applied to the hits, such as a minimum alignment length or percentage identity cutoff?

5)    The confidence score ≥ 0.90 was mentioned for the interactions in the PPI network. Can you provide more information about this score? How is it calculated, and what does it represent in terms of the reliability or strength of the interactions?

6)    Could you explain how the "high-throughput lab experiments" and "curated databases" were utilized to derive the interactions in the PPI network? Which specific databases were included in the analysis, and were there any considerations for the reliability or confidence of the interactions?

Discussion:

1)    How does the expression of pro-inflammatory cytokine IL-1b correlate with antiviral activity – please elaborate.

2)    Why were the genes encoding cytokines not differentially modulated except at 14 days post-infection?

3)    What is the implication of the up-regulation of T cell-specific genes CD4 and CD8a at 14 days post-infection – Please elaborate?

4)    What is the significance of the up-regulation of myosin heavy chain encoding genes in the response to NNV infection at 14 days post-infection?

Minor comments:

1)    Line 32: aquaculture is*

2)    Add a reference for first sentence on Line 33

3)    Line 44: of two positive sense ssRNA

4)    Line 59: well characterized.

5)    Figure 2A: Graph had Percent survival on Y-Axis, whereas text above reads ‘As shown in Figure 2A, the daily mortalities reached a peak on days 5-6 and continued to be high on days 8-12, reaching over 83 % of the total mortality incurred during the experiment.’ 83% survival means 17% mortality – not sure I follow the text. Kindly address!

6)    Most of the graphs have an alphabet, which authors define as follows: ‘Different lower-case letters denote statistically significant differences.’ It would help readers understand what they stand for <0.05; 0.01, etc?

Author Response

Response to Reviewer 1 Comments

Response: We would like to thank the time and effort that reviewer 1 put on the comments of our manuscript. We believe that the corrections/additions based on the reviewers’ comments have greatly improved the quality of our manuscript. Here you will find all original comments and the updated information that was included to clarify our results.

Major comments:

It would help if the authors could elaborate more on these questions below:

Methods/ Point 1: Can you provide more details about the specific criteria used to design the primers for gene expression analysis? How were the potential secondary structures (homo- or cross-dimers, hairpin structures) and primer specificity checked?

Response Μ1: We thank the reviewer for this question. Designing optimal PCR primer sequences is one of the most critical factors for successful PCR with sensitive, specific, and assay-to-assay reproducible results. The process we followed for in silico design and study of primer pairs consists of sequential steps summarized in the following figure. The described procedure was followed to design primer pairs for all genes presented in the manuscript. The first step of the study was to retrieve the reference sequence for each gene from public databases, e.g. NCBI nucleotide database (https://www.ncbi.nlm.nih.gov/nuccore/), which is a collection of sequences from several sources, including GenBank, RefSeq, Third-Party Annotation (TPA) and Protein Data Bank (PDB). Then we performed bibliographic search to find published primers set for the genes of interest, based on the selected reference sequence. Following, we utilized publicly available software for primer designing. The design tools we used were Primer-Blast, IDT_Primer Quest and Primer3Plus. The specific parameters which were set in each software for optimum primer pairs design were: i. PCR product size: 50-200 bp; ii. GC content: 40-60%; iii. Tm: 50-60 °C; iv. primer size: 15-30 bases; v. absence of primer dimers and hairpin structures; v. 3’ end <2GC. All resulted primers sets (both newly designed and previously published) were subjected to further evaluation based on the above mentioned criteria. The primer secondary structures were analysed with the publicly available software Oligo Analyzer (http://eu.idtdna.com/analyzer/Applications/OligoAnalyzer/). Based on the software, Hairpin structures with ΔG > -2 kcal/mol at the 3' end and ΔG > -3 kcal/mol in the middle of the primer (ΔG: Gibbs Free Energy, higher negative ΔG values ​​indicate more stable structures) were considered acceptable. Primer homodimers (i.e. two strands of the same primer interact) with ΔG > -5 kcal/mol at the 3' end and ΔG > -6 kcal/mol inside the primers were acceptable. Primer pair heterodimers (i.e. two primers interact with each other) with ΔG > -5 kcal/mol at the 3' end and ΔG > -6 kcal/mol inside the primers were acceptable. Finally, primers pairs specificity was checked by extensive BLAST analysis. Analysis reports for primer pairs of each gene are available and could be send to the reviewer upon request.

Methods/ Point 2: Could you elaborate on how the minimum information needed for publication of qRT-PCR experiments (MIQE) guidelines were followed?

Response Μ2: By principle, our research group applies the MIQE guidelines for any RT-qPCR involving protocol. The experimental design and all aspects of the procedure are based on the provided check list which can be found in reference [67] for minimum information needed for publication of qRT-PCR experiments (MIQE) guidelines. The specific check list for this study can be found at the end of this response.

Methods/ Point 3: How was the reference gene, b-actin, chosen for this study? Were there any validation experiments conducted to confirm its stability under the experimental conditions?

Response Μ3: B-actin gene has been used broadly as the reference gene for gene expression quantification in D. labrax and other fish species, as seen in the related literature, e.g. [12], [18], [28], [33], [34], [36], [37] and [Tang R, Dodd A, Lai D, McNabb WC, Love DR. Validation of zebrafish (Danio rerio) reference genes for quantitative real-time RT-PCR normalization. Acta Biochim Biophys Sin (Shanghai) 2007, 39:384-390; Ingerslev HC, Pettersen EF, Jakobsen RA, Petersen CB, Wergeland HI. Expression profiling and validation of reference gene candidates in immune relevant tissues and cells from Atlantic salmon (Salmo salar L.). Mol Immunol 2006, 43:1194-1201]. The head-kidney b-actin expression stability in the tested time points, were assessed in a preliminary experiment of 3 biological samples with 3 technical replicated for each sample, and the respective results were: sample 1 Ct: 16.83±0.80; sample 2 Ct: 19.91±1.08; sample3 Ct: 17.15±1.24.

Methods/ Point 4: When performing BLASTX against the NCBI non-redundant (NR) database, what was the e-value threshold used to determine the significance of the hits? Were there any filters applied to the hits, such as a minimum alignment length or percentage identity cutoff?

Response Μ4: Τhe e-value threshold which was used to determine the significance of the hits for BLASTx was set to 1.0E-3. For all others filters, the default values of OmicsBox software were used, i.e. no taxonomy filter was used, the blasted word size was 3, a low complexity elimination filter was used, the highest scoring alignment (HPS) length cut off was set to 33.

Methods/ Point 5: The confidence score ≥ 0.90 was mentioned for the interactions in the PPI network. Can you provide more information about this score? How is it calculated, and what does it represent in terms of the reliability or strength of the interactions?

Response Μ5: As described in “The Search Tool for the Retrieval of Interacting Genes/Proteins database (STRING v11.5)” homepage (https://string-db.org/cgi/info): “In STRING, each protein-protein interaction is annotated with one or more 'scores'. Importantly, these scores do not indicate the strength or the specificity of the interaction. Instead, they are indicators of confidence, i.e. how likely STRING judges an interaction to be true, given the available evidence. All scores rank from 0 to 1, with 1 being the highest possible confidence. A score of 0.5 would indicate that roughly every second interaction might be erroneous (i.e., a false positive).”. Therefore, the confidence score indicates that only 1 of 10 interactions might be a false positive, giving a reliable result without information about interaction strength. The combined confidence scores are computed by integrating the probabilities from the various different types of evidence ('evidence channels'), while correcting for the probability of randomly observing an interaction. A more detailed methodology is described in the following reference, which is added to the manuscript:

[71] von Mering C, Jensen LJ, Snel B, Hooper SD, Krupp M, Foglierini M, Jouffre N, Huynen MA, Bork P. STRING: known and predicted protein-protein associations, integrated and transferred across organisms. Nucleic Acids Res. 2005 Jan 1;33(Database issue): D433-7. doi: 10.1093/nar/gki005. PMID: 15608232; PMCID: PMC539959.

Methods/ Point 6: Could you explain how the "high-throughput lab experiments" and "curated databases" were utilized to derive the interactions in the PPI network? Which specific databases were included in the analysis, and were there any considerations for the reliability or confidence of the interactions?

Response Μ6: The Search Tool for the Retrieval of Interacting Genes/Proteins database (STRING v11.5) is a biological database and web resource of known and predicted protein–protein interactions. STRING imports data from experimentally derived protein–protein interactions through literature curation and store computationally predicted interactions from: (i) text mining of scientific texts, (ii) interactions computed from genomic features, and (iii) interactions transferred from model organisms based on orthology (Wikipedia). Based on these features, a PPI network is predicted with a specific confidence score as described in the above mentioned publication ([71] von Mering et al., 2005). All predicted or imported interactions are benchmarked against a common reference of functional partnership as annotated by KEGG (Kyoto Encyclopedia of Genes and Genomes). As stated in their webpage, the STRING extract experimental/ biochemical data from the following databases: DIP, BioGRID, HPRD, IntAct, MINT and PDB. STRING extract curated data from the following databases: Biocarta, BioCyc, Gene Ontology, KEGG and Reactome. The STRING tool is frequently used and well accepted by scientists in the field, and based on the calculated scores there are no considerations for the reliability of the predicted interactions.

Discussion/ Point 1: How does the expression of pro-inflammatory cytokine IL-1b correlate with antiviral activity – please elaborate

Response D1: As described in reference [44], most in vivo studies have been focused on the transient and local effects of IL-1b on the immune system, establishing the IL-1b role in antibacterial activity. To date, few studies have analysed the direct antiviral activity of fish IL-1b. However, several previous studies have indicated its important role on NNV infection. Therefore, we included the IL-1b study in all time points in an attempt to gain more information on IL-1b antiviral activity.

A recent study (Joo MS, Choi KM, Kang G, Woo WS, Kim KH, Sohn MY, Son HJ, Han HJ, Choi HS, Kim DH, Park CI. Red sea bream interleukin (IL)-1β and IL-8 expression, subcellular localization, and antiviral activity against red sea bream iridovirus (RSIV). Fish Shellfish Immunol. 2022, 128:360-370. doi: 10.1016/j.fsi.2022.07.040) reports that in humans, there have been reports that IL-1b directly induces IFN-γ expression, e.g. by co-stimulating human natural killer (NK) cells to produce IFN-γ in vitro. In addition, IL-1b has been identified as an IL-12-inducing agent, acting conjointly with CD40 ligand (CD40L) on human monocyte-derived dendritic cells (DCs) in vitro. Therefore, co-stimulation with IL-1b and IL-12 may have induced the expression of IFN-γ, which is a well-known inducer of Mx1. The authors have observed the same effect on Pagrus major or red sea bream fin (PMF) cells; i.e IL-1β overexpression induced the expression of interferon gamma (IFN-γ), Mx1, IL-8, IL-10, TNF-α, and MyD88. In parallel, IL-1b also induces itself, which serves as a positive feedback loop to amplify the IL-1 response in an autocrine or paracrine manner. In the same study it was proved, that overexpression of IL-1b and PMF cells significantly reduced viral genome copies of red sea bream iridovirus (RSIV) compared to the control. However, since currently there are only a few studies on the IL-1b antiviral activity, and the scope of the present study was to gain an overview of cytokines antiviral role in a specific time course, as we stated in the discussion section, a more detailed analysis on IL-1b activity should be performed to further elucidate the antiviral mechanism.

Discussion/ Point 2: Why were the genes encoding cytokines not differentially modulated except at 14 days post-infection?

Response D2: Indeed, based on the published references we expected more intense responses on cytokine genes expression. However, as seen in figure 4 the cytokines were slightly modulated at the early time points. Based on our experimental design, the NNV virus was used at concentrations causing low mortality rate (~20%) in order to achieve long term infection of the D. labrax. Also, the D. labrax used belonged in an NNV resistant family [Vela-Avitúa S, Thorland I, Bakopoulos V, Papanna K, Dimitroglou A, Kottaras E, Leonidas P, Guinand B, Tsigenopoulos CS, Aslam ML. Genetic Basis for Resistance Against Viral Nervous Necrosis: GWAS and Potential of Genomic Prediction Explored in Farmed European Sea Bass (Dicentrarchus labrax). Front Genet. 2022; 13:804584. doi: 10.3389/fgene.2022.804584]. Therefore, there is a possibility that the cytokine expression levels were kept low as a result of the host resistance mechanism to overcome the virus infection. That study is currently undergoing by our research groups, with analysis of NNV infection in D. labrax families with different genetic backgrounds (in respect of NNV resistance), but conclusive results aren’t available yet.

Discussion/ Point 3: What is the implication of the up-regulation of T cell-specific genes CD4 and CD8a at 14 days post-infection – Please elaborate.

Response D3: To date, there is lack of information (to the best of our knowledge) on T-cell specific genes expression levels during persistent infection in fish. A recent study (Snell LM, Osokine I, Yamada DH, De la Fuente JR, Elsaesser HJ, Brooks DG. Overcoming CD4 Th1 Cell Fate Restrictions to Sustain Antiviral CD8 T Cells and Control Persistent Virus Infection. Cell Rep. 2016 Sep 20;16(12):3286-3296. doi: 10.1016/j.celrep.2016.08.065.) in mice, reported that: ‘In response to many persistent virus infections, virus-specific T cells are either physically deleted or persist in an attenuated (exhausted) state characterized by a distinct transcriptional program, alterations in antiviral and immune-stimulatory cytokines and a decreased ability to proliferate or lyse virally infected cells. Following a viral infection, CD4 T cells predominantly differentiate into Th1 cells that sustain CD8 T cell responses to kill virus infected cells or develop into T follicular helper (Tfh) cells that mediate B cell differentiation and antibody production. Although CD4 help at the onset of persistent virus infection is initially required to promote the CD8 T cell and antibody responses required for long-term control of infection, the subsets best suited to maintain long-term immunity and control an established persistent virus infection are unclear’. The authors demonstrated that the loss of CD4 Th1 cells underlies the progressive CD8 T cell decline and dysfunction that prevents control of persistent infection, suggesting that restoring CD4 Th1 cells enhances virus-specific CD8 T cell numbers and function facilitating control of the persistent infection. Based on our finding, we assume that the distinct up-regulation of CD4 and CD8 14 dpi reflects the host attempt to overcome the virus effects and maintain the persistent state, avoiding further damage. However, these findings are preliminary and a more detailed analysis should be performed to further elucidate the underlying mechanism. The present comment was incorporated in the manuscript in line 516.

Discussion/ Point 4: What is the significance of the up-regulation of myosin heavy chain encoding genes in the response to NNV infection at 14 days post-infection?

Response D4: As described in the manuscript, myosin participates in immune function regulation with notable role in viral infections, as proved by studies with several which are summarized in the manuscripts discussion section. Several transcriptomic and proteomic studies on NNV infection have highlighted the differential expression of different types of myosin during the infection. To the best of our knowledge, no in depth analysis on myosin role has been published. Therefore, based on the current literature and our findings we assume that myosin up-regulation 14 dpi might be indicative of a shift in virus entry mechanisms to host cells related to the persistent state, a hypothesis which needs further study. However, at the moment we are lacking the data to elucidate the effect of myosin upregulation to NNV infection.

Minor comments:

Point 1: Line 32: aquaculture is*

Response 1: The syntactic error in line 32 was corrected.

Point 2: Add a reference for first sentence on Line 33

Response 2: The following reference was added in line 33 for the first sentence:

Toubanaki, D.K.; Efstathiou, A.; Karagouni, E. Transcriptomic Analysis of Fish Hosts Responses to Nervous Necrosis Virus. Pathogens. 2022, 11(2), 201. doi: 10.3390/pathogens11020201.

Point 3: Line 44: of two positive sense ssRNA

Response 3: The grammar error in line 44 was corrected, as suggested.

Point 4: Line 59: well characterized.

Response 4: The suggested phrase was added in line 59.

Point 5: Figure 2A: Graph had Percent survival on Y-Axis, whereas text above reads ‘As shown in Figure 2A, the daily mortalities reached a peak on days 5-6 and continued to be high on days 8-12, reaching over 83 % of the total mortality incurred during the experiment.’ 83% survival means 17% mortality – not sure I follow the text. Kindly address!

Response 5: We thank the reviewer for that comment. To clarify the sentence was modified as follows “As shown in Figure 2A, the daily mortalities reached a peak on days 5-6 and continued to be high on days 8-12 (as indicated by arrows), reaching over 83 % of the total mortality incurred during the experiment (i.e. 40 fish of 48 fish in total died up to day 12).”

Point 6: Most of the graphs have an alphabet, which authors define as follows: ‘Different lower-case letters denote statistically significant differences.’ It would help readers understand what they stand for <0.05; 0.01, etc?

Response 6: In figures 3 – 7, statistical differences were assessed with one-way ANOVA followed by Tukey multiple comparison test, as stated in their respective legends. The statistically significant differences were depicted using the Compact letter display statistical method instead of using the asterisks method, because the later resulted in very complicated figures. The Compact Letter Display (CLD) is a statistical method to clarify the output of multiple hypothesis testing when using the ANOVA and Tukey's range tests. CLD facilitates the identification of variables, or factors, that have statistically different means (or averages) vs. the ones that do not have statistically different means (or averages). The basic technique of compact letter display is to label variables by one or more letters, so that variables are statistically indistinguishable if and only if they share at least one letter. Each variable that shares a mean that is not statistically different from another one will share the same letter (Wikipedia). Therefore, the use of significance levels (<0.05, etc) it’s not applicable in the selected depiction.

Table 1. MIQE checklist for authors, reviewers, and editors.a

Item to check

Section

Importance

Item to check

Section

Importance

Experimental design

qPCR oligonucleotides

Definition of experimental and control groups

2.1, fig.1

E

Primer sequences

Table S2

E

Number within each group

2.1, fig.1

E

RTPrimerDB identification number

n/a

D

Assay carried out by the core or investigator’s laboratory?

2.1

D

Probe sequences

Table S2

Dd

Acknowledgment of authors’ contributions

Author Contributions

D

Location and identity of any modifications

n/a

E

Sample

Manufacturer of oligonucleotides

Can be provided

D

Description

2.1

E

Purification method

Can be provided

D

Volume/mass of sample processed

n/a

D

qPCR protocol

Microdissection or macrodissection

2.1

E

Complete reaction conditions

2.7

E

Processing procedure

2.1

E

Reaction volume and amount of cDNA/DNA

2.7

E

If frozen, how and how quickly?

2.1

E

Primer, (probe), Mg2+, and dNTP concentrations

2.7

E

If fixed, with what and how quickly?

n/a

E

Polymerase identity and concentration

2.7

E

Sample storage conditions and duration

2.1

E

Buffer/kit identity and manufacturer

2.7

E

Nucleic acid extraction

Exact chemical composition of the buffer

-

D

Procedure and/or instrumentation

2.6

E

Additives (SYBR Green I, DMSO, and so forth)

2.7

E

Name of kit and details of any modifications

2.6

E

Manufacturer of plates/tubes and catalog number

Can be provided

D

Source of additional reagents used

2.6

D

Complete thermocycling parameters

2.7

E

Details of DNase or RNase treatment

n/a

E

Reaction setup (manual/robotic)

-

D

Contamination assessment (DNA or RNA)

2.6

E

Manufacturer of qPCR instrument

2.7

E

Nucleic acid quantification

2.6

E

qPCR validation

Instrument and method

2.6

E

Evidence of optimization (from gradients)

Can be provided

D

Purity (A260/A280)

2.6

D

Specificity (gel, sequence, melt, or digest)

Can be provided

E

Yield

2.6

D

For SYBR Green I, Cq of the NTC

Can be provided

E

RNA integrity: method/instrument

2.8

E

Calibration curves with slope and y intercept

Can be provided

E

RIN/RQI or Cq of 3' and 5' transcripts

2.8

E

PCR efficiency calculated from slope

Table 2

E

Electrophoresis traces

n/a

D

CIs for PCR efficiency or SE

-

D

Inhibition testing (Cq dilutions, spike, or other)

-

E

r2 of calibration curve

Can be provided

E

Reverse transcription

Linear dynamic range

Can be provided

E

Complete reaction conditions

2.6

E

Cq variation at LOD

Can be provided

E

Amount of RNA and reaction volume

2.6

E

CIs throughout range

-

D

Priming oligonucleotide (if using GSP) and concentration

2.6

E

Evidence for LOD

Can be provided

E

Reverse transcriptase and concentration

2.6

E

If multiplex, efficiency and LOD of each assay

n/a

E

Temperature and time

2.6

E

Data analysis

Manufacturer of reagents and catalogue numbers

2.6

D

qPCR analysis program (source, version)

2.7

E

Cqs with and without reverse transcription

Can be provided

Dc

Method of Cq determination

2.7

E

Storage conditions of cDNA

-

D

Outlier identification and disposition

n/a

E

qPCR target information

Results for NTCs

Can be provided

E

Gene symbol

Table S2

E

Justification of number and choice of reference genes

2.7

E

Sequence accession number

Table S2

E

Description of normalization method

2.7

E

Location of amplicon

-

D

Number and concordance of biological replicates

2.7

D

Amplicon length

Table S2

E

Number and stage (reverse transcription or qPCR) of technical replicates

2.7

E

In silico specificity screen (BLAST, and so on)

Can be provided

E

Repeatability (intraassay variation)

Can be provided

E

Pseudogenes, retropseudogenes, or other homologs?

n/a

D

Reproducibility (interassay variation, CV)

Figures 3-7

D

Sequence alignment

Can be provided

D

Power analysis

n/a

D

Secondary structure analysis of amplicon

Can be provided

D

Statistical methods for results significance

Fig. 3-7  legends

E

Location of each primer by exon or intron (if applicable)

Can be provided

E

Software (source, version)

Can be provided

E

What splice variants are targeted?

n/a

E

Cq or raw data submission with RDML

-

D

a All essential information (E) must be submitted with the manuscript. Desirable information (D) should be submitted if available. If primers are from RTPrimerDB, information on qPCR target, oligonucleotides, protocols, and validation is available from that source.

b FFPE, formalin-fixed, paraffin-embedded; RIN, RNA integrity number; RQI, RNA quality indicator; GSP, gene-specific priming; dNTP, deoxynucleoside triphosphate.

c Assessing the absence of DNA with a no–reverse transcription assay is essential when first extracting RNA. Once the sample has been validated as DNA free, inclusion of a no–reverse transcription control is desirable but no longer essential.

d Disclosure of the probe sequence is highly desirable and strongly encouraged; however, because not all vendors of commercial predesigned assays provide this information, it cannot be an essential requirement. Use of such assays is discouraged.

Reviewer 2 Report

Comments and Suggestions for Authors

This manuscript, by Toubanaki et al, reported how Nervous Necrosis Virus (NNV) infection affects selected immnue system-related gene expression and the overall transcriptome in European sea bass. Over 200 sea bass were challenged with NNV infection, and head kidney samples were collected at several time points post infection. The mRNA expression level of certain genes was measured by RT-qPCR during a 28-day period, while significantly altered gene expression was observed for several genes at 14 dpi. A high-throughput transcriptomic analysis was also performed for head kidney samples collected at 14 dpi. Overall, this manuscript presented important information of gene expression regulation of NNV infected sea bass at the transcriptomic level, which also provided insights of NNV persistent infection. 

The specific minor comments are:

1. line 23, "at this time-point": it is better to say "at 14 dpi" here to make it clearer. 

2. I wonder how samples were randomly collected when there were deceased fish? Were the deceased fish removed from the random pool and excluded?

3. It is worthy mentioning one limitation of this study in the discussion that the protein translation level is not tested. A simple example will be Figure 7A and B where the mRNA level doesn't always match with the protein expression level. 

4. The font size of several figures is too small, such as Figure 4, 5, 6 and 7.

5. Line 384-407: Many information of this paragraph is unrelated to the current study. Please consider simplifying by deleting some detailed descriptions of unrelated pathways.

6. It is recommended that a supplementary spreadsheet data file can be provided listing necessary raw data of the transcriptome analysis. This could be an important source for future work in this field. 

Author Response

Response to Reviewer 2 Comments

We would like to thank reviewer 2 for the time and effort put on the comments of our manuscript. We believe that the corrections/additions based on the reviewers’ comments have greatly improved the quality of our manuscript. Here you will find all original comments and the updated information that was included to clarify our results.

The specific minor comments are:

Point 1: line 23, "at this time-point": it is better to say "at 14 dpi" here to make it clearer.

Response 1: the proposed change was incorporated in the manuscript.

Point 2: I wonder how samples were randomly collected when there were deceased fish? Were the deceased fish removed from the random pool and excluded?

Response 2: We thank the reviewer for this comment. At each time point the deceased fish were removed, and the samples were randomly selected from the remaining live fish, since our goal was to study the immune responses of the diseased/ survivor/ carrier fish. To clarify this point, we modified the phrase in lines 672-673 as follows: “At each specific time point, the deceased fish were removed, and 15 fish of the remaining live population (5 samples from each tank) were randomly selected”.

Point 3: It is worthy mentioning one limitation of this study in the discussion that the protein translation level is not tested. A simple example will be Figure 7A and B where the mRNA level doesn't always match with the protein expression level.

Response 3: To emphasize on this aspect, the following text was added in lines 530-534: “It should be noted, that the present study was focused on immune related gene-expression analysis which offers a timeshot depiction at the transcription level. The expressed protein levels may be different, as we observe in figure 7 for IgHM. Therefore, the assessment of the respective protein levels in a future study will be beneficial for immune responses under-standing in depth.”

Point 4: The font size of several figures is too small, such as Figure 4, 5, 6 and 7.

Response 4: Fonts in figures 3, 4, 5, 6, 7 were increased as suggested.

Point 5: Line 384-407: Many information of this paragraph is unrelated to the current study. Please consider simplifying by deleting some detailed descriptions of unrelated pathways

Response 5: Following the reviewer 2 suggestion, descriptions of unrelated pathways were removed.

Point 6: It is recommended that a supplementary spreadsheet data file can be provided listing necessary raw data of the transcriptome analysis. This could be an important source for future work in this field.

Response 6: Following the reviewer 2 recommendation, an excel data file containing data related to transcriptome analysis has been included in supplementary material as Table S3.